# Data-driven recombination detection in viral genomes

Tommaso Alfonsi [1,3], Anna Bernasconi [1,3] ✉, Matteo Chiara [2] & Stefano Ceri [1]

Recombination is a key molecular mechanism for the evolution and adaptation of viruses. The first recombinant SARS-CoV-2 genomes were recognized in 2021; as of today, more than ninety SARS-CoV-2 lineages are designated as recombinant. In the wake of the COVID-19 pandemic, several methods for detecting recombination in SARS-CoV-2 have been proposed; however, none could faithfully confirm manual analyses by experts in the field. We hereby present RecombinHunt, an original data-driven method for the identification of recombinant genomes, capable of recognizing recombinant SARS-CoV-2 genomes (or lineages) with one or two breakpoints with high accuracy and within reduced turn-around times. ReconbinHunt shows high specificity and sensitivity, compares favorably with other state-of-the-art methods, and faithfully confirms manual analyses by experts. RecombinHunt identifies recombinant viral genomes from the recent monkeypox epidemic in high concordance with manually curated analyses by experts, suggesting that our approach is robust and can be applied to any epidemic/pandemic virus.

Recombination is a key molecular mechanism used by RNA viruses to boost their evolution. Recombination occurs both in viruses with segmented and non-segmented genomes; parental strains to a recombinant virus are referred to as "donor" and "acceptor." Recombination requires co-circulation and co-infection in the same host; the clinical and epidemiological relevance is substantial since recombinant viral strains have been associated with altered viral host tropism, enhanced virulence, host immune evasion, and the development of resistance to antivirals[1,2]. In light of these considerations, and in hindsight from the recent global scale COVID-19 epidemic, the need for the development of novel and rapid methods to identify recombination has been increasingly recognized by international health authorities and researches[3,4]. Phylogenetic analyses are essential to monitoring the spread and evolution of viruses[5]. All phylogeny-based approaches assume that the shared history of pathogens, isolated from different hosts, can be described by a branching phylogenetic tree. Recombination breaks this assumption and impacts the application of phylogenetic methods for the reconstruction of chains of

contagion, viral evolution, and ultimately genomic surveillance of pathogens[6,7].

During the COVID-19 pandemic SARS-CoV-2 has accumulated over 130K distinct nucleotide mutations, leading to the emergence of more than 2K lineages. In the first three years of the COVID-19 pandemic, limited levels of recombination were observed, although an increased number of recombinant lineages has been reported since the emergence and spread of novel variants of concern (VOC)[8]. Indeed, as SARS-CoV-2 evolved and differentiated, several recombination events have been recognized, highlighting once more the importance of recombination as a molecular mechanism for the generation of genomic and phenotypic diversity in epidemic/pandemic viruses[9-16].

For instance, as the Omicron SARS-CoV-2 variant became dominant worldwide, about 60 recombinant lineages have been identified only within Omicron. XE (also known as V-22APR-02 in Public Health England) was considered the most concerning Omicron lineage during 2022, given a reported growth advantage[17]. Most Variants Of Interest and Variants Under Monitoring according to the World Health

[1]Department of Electronics, Information, and Bioengineering, Politecnico di Milano, Via Ponzio 34/5, 20133 Milan, Italy. [2]Department of Biosciences, Università degli Studi di Milano, Via Celoria 26, 20133 Milan, Italy. [3]These authors contributed equally: Tommaso Alfonsi, Anna Bernasconi. ✉e-mail: anna.bernasconi@polimi.it

Organization – at the time of writing[18] – are descendants of the XBB recombinant lineage.

The rapid emergence of novel SARS-CoV-2 viral lineages and their potential epidemiological implications has called for continuous monitoring of viral genome evolution in the last few years. The largest available collection of genomes has been curated by GISAID[19], which reached 15.2 million deposited viral sequences in April 2023. Genomic surveillance efforts initially focused on the monitoring of single amino acid changes. As the COVID-19 pandemic progressed, research interests shifted toward the study of mutational signatures and variants associated with increased transmission rates and reduced antigenicity, and possibly hampering testing, treatment, and vaccine development[20–22]. A number of methods were proposed to allow automatic early detection of variants[23–27]. Instead, interest in the automatic identification of recombination in SARS-CoV-2 started at a later stage.

Recombination in viral genomes is often identified using algorithms implemented in programs such as Simplot[28]; GARD[29]; 3SEQ[30] and its improved version[31]; RDP3 (Recombination Detection Program version 3[32], including four previously proposed tools) and its extensions RDP4[33] / RDP5[34]; and RAPR[35] (Recombination Analysis PRogram). In short, these methods apply phylogenetic-based approaches to identify recombination hotspots and pinpoint patterns of interest with matrix-based visualizations. However, none of these methods was specifically devised to analyze/deal with big data/millions of genome sequences. Moreover, some of these algorithms account for all polymorphic sites equally, regardless of phylogenetic information, and hence might be prone to systematic errors if/when applied to a sparse, arbitrary selection of genome sequences.

Some studies already applied the methods discussed above to SARS-CoV-2. For instance, Lytras et al.[36] used GARD for identifying recombination hotspot in proximal SARS-CoV-2 ancestors; Pollett et al.[37] employed RDP4 on 100K sequences (dataset of August 2021); 3SEQ was applied by Boni et al.[38] and Jackson et al.[39] for analysis of mosaic signals and breakpoint identification; while Shiraz and Tripathi[8] combined 3SEQ and RDP5 for assigning parent lineages.

Ignatieva et al.[40] proposed KwARG, a parsimony-based method to reconstruct possible genealogical histories of SARS-CoV-2 and disentangle recombination based on a statistical framework. Similar to other comparable works[39,41], the method however suffers from a limited resolution and can not fully resolve pairs of recombinant donors/ acceptor sequences at the lineage level.

Zhou et al.[42] introduced VirusRecom, an algorithm that uses information theory to infer recombination. The method is applied only to simulated data and a limited selection of recombinant lineages (XD, XE, and XF); these settings cannot be considered a comprehensive evaluation.

Also due to these limitations, during the COVID-19 pandemic, the hunt for recombinant SARS-CoV-2 viral lineages was performed mostly manually by experts who reported evidence on the Pango designation GitHub repository in the form of issues; these issues are broadly documented and allow for discussion with other peers[43].

Only the RIPPLES method[6] and related RIVET software[44] have been applied to the complete collection of SARS-CoV-2 genome sequences.

Here, we present RecombinHunt, a new automatic method (see Fig. 1) for effectively and efficiently detecting recombinations by analyzing the complete data corpus of SARS-CoV-2. Our approach relies exclusively on data-driven methods; the method produces results that are easily inspectable on simple visual reports (see Supplementary Notes 1–4). RecombinHunt's conceptual framework stems from a long-lasting tradition of statistical methods for the detection of intragenic recombination (started with Stephens[45]) - and is, in a way, related to substitution distribution models[46] - but differs in several key aspects. First, RecombinHunt does not implement a triplet-based approach (such as RDP and 3SEQ) - where every candidate recombinant sequence is evaluated by extensive comparisons with all the potential pairs of parents - but instead, it abstracts independent clusters of genomes as defined by a reference classification system/nomenclature in the form of a list of characterizing mutations. Subsequently, every candidate recombinant sequence is assessed by computing its similarity/dissimilarity with many existing lineages/groups of similar genome sequences. Second, while previously established methods (such as LARD[47]) employ likelihood-based approaches to infer the most probable phylogenetic model and derive the evolutionary origin of a sequence (no-recombination, recombination, number of breakpoints), RecombinHunt does not reconstruct phylogenies but computes the likelihood of a collection of pre-defined designations/lineages and their combinations (recombinants) based on the mutations in the target sequence. Third, although RecombinHunt identifies the most-likely candidate parents for a recombinant sequence by using an algorithm conceptually similar to the hypergeometric random walk described in Boni et al.[30] and Lam et al.[31], unlike RecombinHunt these methods do not explicitly account for the frequency of each distinct point mutation, and are thereby bound to a completely different statistical framework.

By performing extensive analyses on simulated and publicly available data, the main results of this study include: an accurate assessment of the sensitivity, specificity, and minimum requirements for the application of RecombinHunt; the analysis of lineages designated as recombinant at the time of writing, both at the consensus sequence level and through the analysis of single high-quality sequences; a comparison with RIPPLES –the main competing approach– and the identification of recombinant genomes from the recent monkeypox epidemic.

## Results
### Data collection
We considered 15,271,031 SARS-CoV-2 genomes, downloaded from the GISAID database[19] on April 1st, 2023. Genome sequences were aligned to the SARS-CoV-2 reference genome and nucleotide mutations were identified by the HaploCoV pipeline[27]. To mitigate the impacts of sequencing and assembly errors, genome sequences of uncertain/low quality were excluded (see Methods). We then retained 5,255,228 viral genomes, for which we only considered the assigned Pango lineage and the list of mutations. Overall, a total of 2345 distinct lineages were represented; this dataset includes a total of 57 distinct recombinant lineages with at least one high-quality sequence (denoted by the initial letter 'X' according to the Pango convention for labeling recombinant lineages).

### Approach
Mutation frequencies were estimated across the complete collection of Pango lineages and in the complete collection of SARS-CoV-2 genome sequences. For every lineage in the SARS-CoV-2 reference nomenclature, mutations with a frequency above a 75% threshold were called *characteristic mutations* (see Methods); the list of characteristic mutations for a lineage is denoted as the *lineage mutations-space* in RecombinHunt.

RecombinHunt accepts as input a target genome sequence, in the form of a list of nucleotide mutations. Genome positions with a mutation in the target are denoted as the *target mutations-space*. Candidate "donor" and "acceptor" lineages are defined based on the counts of their mutations in the target mutations-space; we denote as "donor" the lineage with the higher count. For every lineage, the union of the lineage and target mutations-space is denoted as *extended target space*. A cumulative likelihood ratio score is derived according to the following procedure: at each position of the extended target space, we compute the logarithmic ratio between the frequency of the mutation in the lineage and in the complete collection of SARS-CoV-2 genomes. This score is added if the mutation is shared by both the target and the

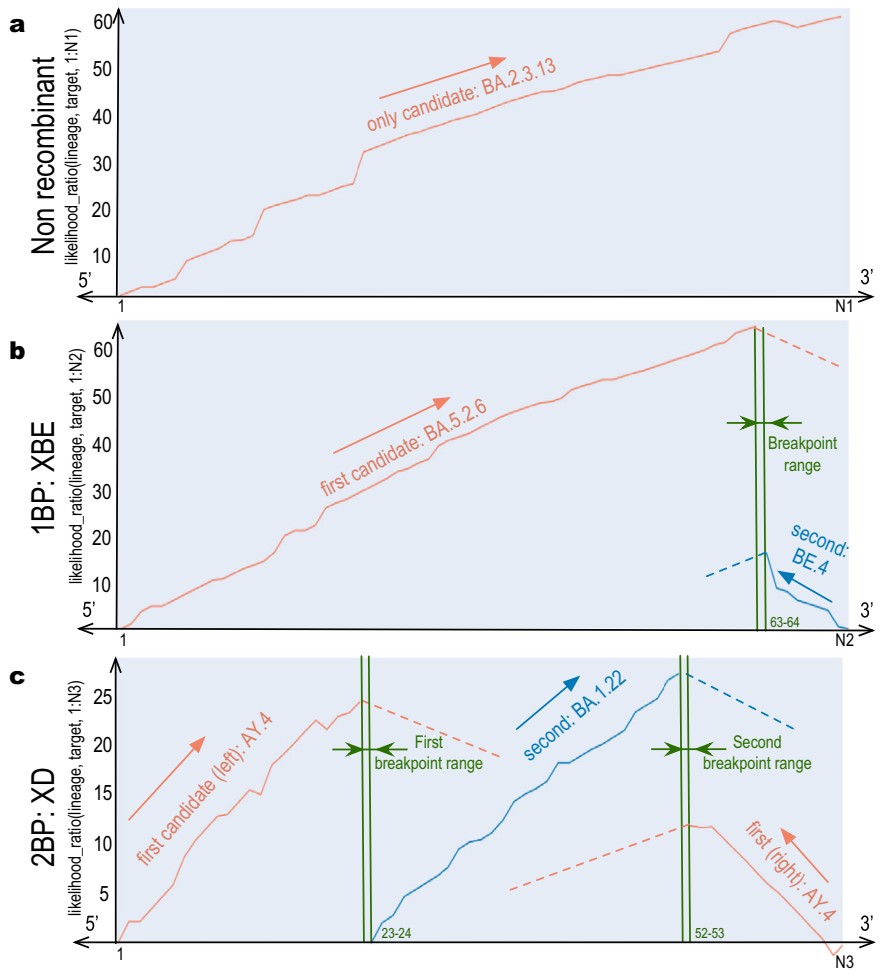

**Fig. 1 | RecombinHunt has three possible outcomes: no recombination, 1 breakpoint recombination, or 2 breakpoints recombination. a** Example of likelihood ratio profile for a non-recombined genome with N1 = 66 mutations, only featuring one donor lineage corresponding to BA.2.3.13. **b** Example of likelihood ratio profile for a recombined genome assigned to XBE Pango lineage with N2 = 72 mutations, breakpoint at the 63rd mutation, donor lineage BA.5.2.6 (from 5'-end to 63rd mutation), and acceptor lineage BE.4 (from 64th mutation to 3'-end). **c** Example of likelihood ratio profile for a recombined genome assigned to XD Pango lineage with N3 = 67 mutations, two breakpoints at the 23rd and 52nd mutations, donor lineage AY.4 (from 5'-end to 23rd mutation and from 53rd mutation to 3'-end), and acceptor lineage BA.1.22 (from 24th to 52nd mutation).

lineage, whereas it is subtracted if the mutation is observed in the lineage but not in the target.

The workflow is represented in Fig. 2. For a target input sequence, likelihood ratio values are computed for all possible lineages; the lineage L1 associated with the maximum value is assigned to the target. If L1 mutations-space differs from the target mutations-space at most in two positions, then the non-recombinant model (Fig. 1a) is selected, and the target is assigned d to L1; else, L1 is designated as the candidate donor. Note that L1 covers the majority of the mutations of the target, located in the genome segment that starts from one of the two ends (either 5' or 3') – denoted as L1's end – and reaches its maximum value at a position designated as max-L1. Upon the identification of a candidate donor, the one-breakpoint model (1BP, Fig. 1b) and two-breakpoint model (2BP, Fig. 1c) are compared in parallel.

In the 1BP model, we search for a lineage L2, starting at the opposite end of the genome – denoted as $L1^{opp}$ end. We consider the L2 lineage associated with the maximum likelihood ratio value (max-L2); if such lineage is not different from L1 in at least three mutations, we recede to the non-recombination case, else we designate it as a candidate acceptor. The interval between coordinates max-L1 and max-L2, where the donor and acceptor lineages reach their maximum likelihood ratio, defines the 'breakpoint range', which is then reduced to a single position (see Methods). An example of a 1BP use case is shown in

Fig. 1b, where a target sequence assigned to the Pango XBE lineage (Pango issue #1246) is correctly recognized to originate from a BA.5.2.6 donor (with 63 mutations) and a BE.4 acceptor (with 9 mutations).

In the 2BP model, the candidate donor L1 lineage is assigned to both ends of the genome; this case is explored only when the target sequence has at least three mutations of L1 at the $L1^{opp}$ end of the genome. We designate as $L1^{opp}$ the portion of the genome between the $L1^{opp}$ end and the point where $L1^{opp}$'s likelihood ratio is maximum, denoted as max-$L1^{opp}$. A candidate acceptor L2 lineage is searched in the space between max-$L1^{opp}$ and max-L1; the lineage L2 with the maximum likelihood ratio is selected, yielding to a breakpoint range, positioned either between max-L1 and max-L2 or between max-$L1^{opp}$ and max-L2; if such breakpoint range is greater than one mutation, it is reduced to a single mutation. In Fig. 1c a target sequence assigned to the XD lineage (according to the issue #444 of Pango) showcases an application of the 2BP model. The target was correctly recognized to be a mosaic of the AY.4 lineage (23 and 15 mutations at both ends) and the BA.1.22 lineage (29 mutations).

The Akaike Information Criterion (AIC)[48] is used to compare the likelihood ratio of 1BP and 2BP models and select the model with the highest likelihood. A further comparison with a "non-recombinant" model is made to cross-check the consistency of the results.

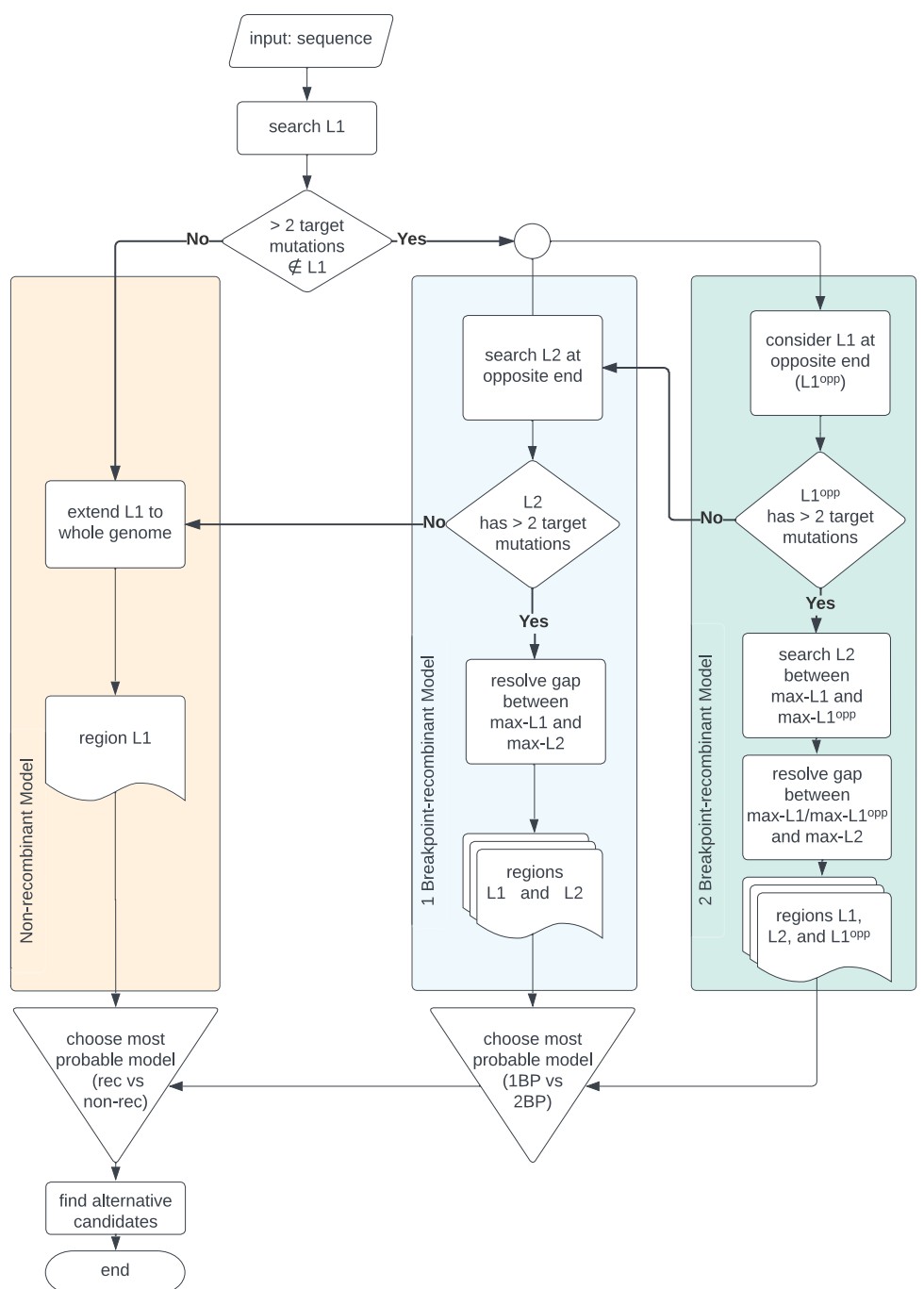

**Fig. 2 | Overall RecombinHunt workflow.** An input viral sequence is considered. The donor lineage is searched based on the cumulative likelihood ratio. Then, three branches are considered: non-recombinant model, one-breakpoint recombination model, and two-breakpoint recombination model. The preferred model is chosen using statistical testing, based on the Akaike information criterion.

As illustrated in Fig. 3, RecombinHunt can identify and resolve alternative combinations of donor/acceptor lineages within phylogenetic clades. Given the best candidates for L1 and L2 (reported in the first row of Fig. 3's tables), alternative candidates must fulfill the following conditions: 1) have a limited difference in likelihood ratio scores compared with the best L1/L2 candidate (AIC $p$-value $\geq 10^{-5}$); 2) reach max-L1 (or max-L2) within a one-mutation distance in the mutations-space from the best candidate; and 3) belong to the same phylogenetic sub-tree as the best candidate. In the case of XBE (Fig. 3a), two candidate lineages i.e., BA.5.2.6 and CP.3, are assigned to the 5′-end of the genome whereas six equivalent candidates are identified for the 3′-end portion. In the case of XD, instead, no alternative candidate donors and acceptors are recovered (Fig. 3b).

### Sensitivity, specificity, minimum requirements

Extensive simulations (see simulated sequences in Supplementary Data 1) were performed to measure the sensitivity and specificity of RecombinHunt and the minimum requirements for its application.

**Sensitivity.** For testing the sensitivity, we considered two SARS-CoV-2 lineages (BA.2 and AY.45) and generated recombinant sequences with one or two breakpoints. Similarly to Turakhia et al.[6], we simulated two

**a**

| RecombinHunt run: 63 sequences labelled with Pango lineage XBE represented by 75%-consensus (72 mutations) | Ground truth (Pango designation issue #1246): BA.5.2* + BE.4.1 Break point 31-53 (of 72) |

**b**

| RecombinHunt run: 14 sequences labelled with Pango lineage XD represented by 75%-consensus (67 mutations) | Ground truth (Pango designation issue #444): B.1.617.2* + BA.1* + B.1.617.2* Break points 23–26 & 51-53 (of 67) |

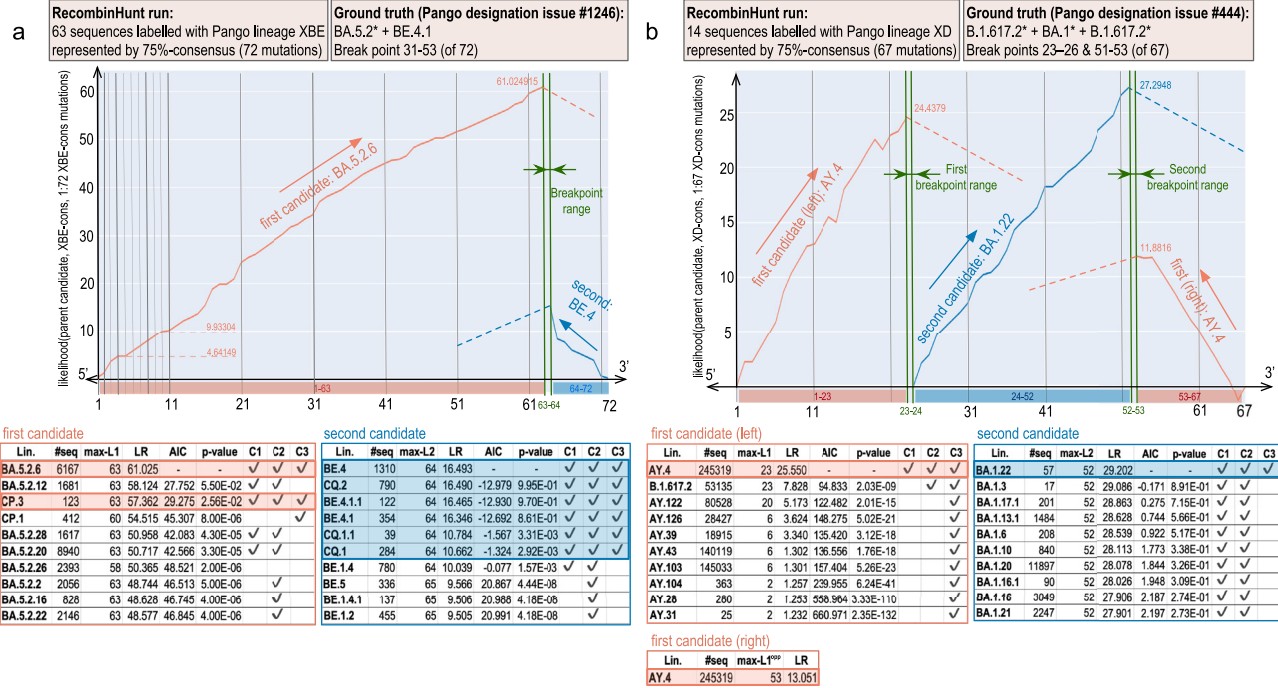

**a — first candidate**

| Lin. | #seq | max-L1 | LR | AIC | p-value | C1 | C2 | C3 |
|---|---|---|---|---|---|---|---|---|
| BA.5.2.6 | 6167 | 63 | 61.025 | - | - | ✓ | ✓ | ✓ |
| BA.5.2.12 | 1681 | 63 | 58.124 | 27.752 | 5.50E-02 | ✓ | | |
| CP.3 | 123 | 63 | 57.362 | 29.275 | 2.56E-02 | ✓ | ✓ | ✓ |
| CP.1 | 412 | 60 | 54.515 | 45.307 | 8.00E-06 | | | ✓ |
| BA.5.2.28 | 1617 | 63 | 50.958 | 42.083 | 4.30E-05 | ✓ | ✓ | |
| BA.5.2.20 | 8940 | 63 | 50.717 | 42.566 | 3.30E-05 | ✓ | ✓ | |
| BA.5.2.26 | 2393 | 58 | 50.365 | 48.521 | 2.00E-06 | | | |
| BA.5.2.2 | 2056 | 63 | 48.744 | 46.513 | 5.00E-06 | | ✓ | |
| BA.5.2.16 | 828 | 63 | 48.628 | 46.745 | 4.00E-06 | | ✓ | |
| BA.5.2.22 | 2146 | 63 | 48.577 | 46.845 | 4.00E-06 | | ✓ | |

**a — second candidate**

| Lin. | #seq | max-L2 | LR | AIC | p-value | C1 | C2 | C3 |
|---|---|---|---|---|---|---|---|---|
| BE.4 | 1310 | 64 | 16.493 | - | - | ✓ | ✓ | ✓ |
| CQ.2 | 790 | 64 | 16.490 | -12.979 | 9.95E-01 | ✓ | ✓ | ✓ |
| BE.4.1.1 | 122 | 64 | 16.465 | -12.930 | 9.70E-01 | ✓ | ✓ | ✓ |
| BE.4.1 | 354 | 64 | 16.346 | -12.692 | 8.61E-01 | ✓ | ✓ | ✓ |
| CQ.1.1 | 39 | 64 | 10.784 | -1.567 | 3.31E-03 | ✓ | ✓ | ✓ |
| CQ.1 | 284 | 64 | 10.662 | -1.324 | 2.92E-03 | ✓ | ✓ | ✓ |
| BE.1.4 | 780 | 64 | 10.039 | -0.077 | 1.57E-03 | ✓ | ✓ | |
| BE.5 | 336 | 65 | 9.566 | 20.867 | 4.44E-08 | | ✓ | |
| BE.1.4.1 | 137 | 65 | 9.506 | 20.988 | 4.18E-06 | | ✓ | |
| BE.1.2 | 455 | 65 | 9.505 | 20.991 | 4.18E-08 | | ✓ | |

**b — first candidate (left)**

| Lin. | #seq | max-L1 | LR | AIC | p-value | C1 | C2 | C3 |
|---|---|---|---|---|---|---|---|---|
| AY.4 | 245319 | 23 | 25.550 | - | - | ✓ | ✓ | ✓ |
| B.1.617.2 | 53135 | 23 | 7.828 | 94.833 | 2.03E-09 | ✓ | | |
| AY.122 | 80528 | 20 | 5.173 | 122.482 | 2.01E-15 | | | ✓ |
| AY.126 | 28427 | 6 | 3.624 | 148.275 | 5.02E-21 | | | ✓ |
| AY.39 | 18915 | 6 | 3.340 | 135.420 | 3.12E-18 | | | ✓ |
| AY.43 | 140119 | 6 | 1.302 | 136.556 | 1.76E-18 | | | ✓ |
| AY.103 | 145033 | 6 | 1.301 | 157.404 | 5.26E-23 | | | ✓ |
| AY.104 | 363 | 2 | 1.257 | 239.955 | 6.24E-41 | | | ✓ |
| AY.28 | 280 | 2 | 1.253 | 558.964 | 3.33E-110 | | | ✓ |
| AY.31 | 25 | 2 | 1.232 | 660.971 | 2.35E-132 | | | ✓ |

**b — second candidate (right)**

| Lin. | #seq | max-L2 | LR | AIC | p-value | C1 | C2 | C3 |
|---|---|---|---|---|---|---|---|---|
| BA.1.22 | 57 | 52 | 29.202 | - | - | ✓ | ✓ | |
| BA.1.3 | 17 | 52 | 29.086 | -0.171 | 8.91E-01 | ✓ | ✓ | |
| BA.1.17.1 | 201 | 52 | 28.863 | 0.275 | 7.15E-01 | ✓ | ✓ | |
| BA.1.13.1 | 1484 | 52 | 28.628 | 0.744 | 5.66E-01 | ✓ | ✓ | |
| BA.1.6 | 208 | 52 | 28.539 | 0.922 | 5.17E-01 | ✓ | ✓ | |
| BA.1.10 | 840 | 52 | 28.113 | 1.773 | 3.38E-01 | ✓ | ✓ | |
| BA.1.20 | 11897 | 52 | 28.078 | 1.844 | 3.26E-01 | ✓ | ✓ | |
| BA.1.16.1 | 90 | 52 | 28.026 | 1.948 | 3.09E-01 | ✓ | ✓ | |
| BA.1.16 | 3049 | 52 | 27.906 | 2.187 | 2.74E-01 | ✓ | ✓ | |
| BA.1.21 | 2247 | 52 | 27.901 | 2.197 | 2.73E-01 | ✓ | ✓ | |

**b — first candidate (right)**

| Lin. | #seq | max-L1$^{ggg}$ | LR |
|---|---|---|---|
| AY.4 | 245319 | 53 | 13.051 |

**Fig. 3 | RecombinHunt recognizes recombination events in one and two-breakpoint cases. a** Search result on 75% consensus-genomes of 63 high-quality sequences assigned to XBE Pango lineage. RecombinHunt selects BA.5.2.6 (child of BA.5.2, see ground truth) as L1 candidate, starting from the 5'-end of the genome; maximum likelihood ratio (LR) 61.025 is reached at mutation 63 (max-L1). Then, RecombinHunt selects BE.4 as L2 candidate in positions (64-72). BA.5.2.6 (left table) and BE.4 (right table) are compared with the following candidate lineages, ranked by their maximum likelihood ratios. Tables report the number of sequences, breakpoint, maximum likelihood ratio. The next columns illustrate the comparison of the candidate with the first of the table: value of one-sided AIC comparison between recombination model and non-recombination model (lower values are when row candidate is similar to the first candidate); p-value of AIC -- without multiple comparison corrections; and three conditions: (C1) marked if p-value is ≥$10^{-5}$; (C2) marked if row breakpoint is at most one mutation apart from the one of the first candidate; (C3) marked if candidate belongs to the same phylogenetic branch as the first one. Candidates with three marks are incorporated into groups, resulting in BA.5.2.6 and CP.3 (candidates for L1) and BE.4, CQ.2, BE.4.1.1, BE.4.1, CQ.1.1, CQ.1 (for L2). **b** Search result on 75% consensus-genome of 14 high-quality sequences assigned to XD Pango lineage. For the 5'-end portion of the genome, RecombinHunt selects AY.4 as L1 candidate (child of B.1.617.2 in the ground truth), with maximum likelihood ratio at the 23rd mutation; AY.4 is also selected for the 3'-end portion of the genome with maximum likelihood ratio at the 53rd position. Then, RecombinHunt identifies the first L2 candidate BA.1.22 (consistent with the ground truth BA.1*) in the 24-52 mutation interval. AY.4 (left table) and BA.1.22 (right table) are compared with the following candidates; no one conjunctively meets the three conditions.

sets of 3500 recombinant sequences (1BP and 2BP cases) each partitioned into seven groups of 500 sequences, and injected increasing levels of noise, by adding -respectively to each group- 0, 3, 5, 10, 15, 20, or 30 mutations out of 4,983 that are non-characteristic and with frequency ≥1/$10^5$ in both parent lineages, at random genomic positions. Note that the generated SARS-CoV-2 genome sequences carry about 60 mutations compared to the reference Wuhan1 genome; hence, adding 3 mutations corresponds to inserting 5% noise, whereas 30 mutations to inserting 50% noise. RecombinHunt achieved an almost perfect sensitivity when the number of added mutations was ≤ 10 (100%-99.4% for 1BP and 99%-95.6% for 2BP). By adding up to 30 mutations, performance slightly decreases as expected; see Table 1a. The breakpoint position was identified correctly (i.e., within a single mutation range) in 99.4% of the simulated 1BP cases and in 97.6% of the simulated 2BP cases.

**Specificity.** We analyzed the false positive rate by evaluating a collection of 3500 randomly selected sequences assigned to the non-recombinant SARS-CoV-2 lineage BA.2. The set was partitioned into seven groups of 500 sequences, and added -respectively to each group- 1, 3, 5, 10, 15, 20, or 30 mutations out of 24,277 that are non-characteristic and with frequency ≥1/$10^5$ in BA.2, at random genomic positions. RecombinHunt classified as non-recombinant -and assigned the correct lineage to- the great majority of the sequences. False positives' ratios ranged from 0.6% to 1.2% to 8.8% for 1, 10 or 30 added mutations; see Table 1b.

**Minimum requirements.** Ideally, RecombinHunt can be applied to any virus for which an adequately large collection of viral genome sequences and a structured classification system (clades or lineages) are available; in these settings, we denote as characteristic mutations of each class those showing a frequency above a given threshold. Here, we estimate the minimum number of sequences necessary to derive a stable set of characteristic mutations. To be characteristic of a lineage $L$, in our model, a mutation $M$ must have a frequency $f$ ≥ threshold above a user-defined value. The minimum number $n$ of genomic sequences required to determine whether the observed frequency of $M$ is above $f$ with a certain level of confidence can be approximated by a Fisher test of a stochastic variable $X \sim Bin(n, f)$ describing the number of independent observations of $M$ in a set of $n$ genome sequences of L; our null hypothesis is then $H_0 = X \geq n^*$threshold $= X \geq K$. Thus, $M$ is characteristic of $L$ if observed at least $K$ times. Based on these assumptions, depending on the threshold and the frequency of $M$ in $L$, we can compute the minimum value of $n$ required to identify a characteristic mutation with a level of confidence of choice (p-value or test acceptance threshold).

RecombinHunt uses a 75% threshold for SARS-CoV-2. If we assume that $M$ has $f = 0.9$, then it descends that $M$ can be considered characteristic of $L$ with a 95% confidence by considering just 16 sequences. Instead, assuming that $M$ has $f = 0.15$ and a characterization threshold of 9%, at least 93 sequences would be required to reach the correct characterization with a confidence ≥95%.

**Table 1 | a Sensitivity analysis**

**(a)**

| Truth | RH prediction | # added random mutations | | | | | | |
|---|---|---|---|---|---|---|---|---|
| | | 0 | 3 | 5 | 10 | 15 | 20 | 30 |
| 1BP | 1BP | 500 (100%) | 499 (99.8%) | 499 (99.8%) | 497 (99.4%) | 485 (97%) | 484 (96.8%) | 464 (92.8%) |
| 1BP | 2BP | – | – | 1 (0.2%) | 1 (0.2%) | 3 (0.6%) | 8 (1.6%) | 21 (4.2%) |
| 1BP | 0BP | – | 1 (0.2%) | – | 2 (0.4%) | 12 (2.4%) | 8 (1.6%) | 15 (3%) |
| 2BP | 2BP | 495 (99%) | 490 (98%) | 490 (98%) | 478 (95.6%) | 467 (93.4%) | 430 (86%) | 394 (78.8%) |
| 2BP | 1BP | – | 2 (0.4%) | 1 (0.2%) | 1 (0.2%) | 3 (0.6%) | 5 (1%) | 16 (3.2%) |
| 2BP | 0BP | 5 (1%) | 8 (1.6%) | 9 (1.8%) | 21 (4.2%) | 30 (6%) | 65 (13%) | 90 (18%) |

**(b)**

| | # added random mutations | | | | | | |
|---|---|---|---|---|---|---|---|
| | 1 | 3 | 5 | 10 | 15 | 20 | 30 |
| False positives rate | 3 (0.6%) | 5 (1%) | 3 (0.6%) | 6 (1.2%) | 20 (4%) | 26 (5.2%) | 44 (8.8%) |

**(c)**

| | minimum # sequences | | |
|---|---|---|---|
| Mutations-noise levels | SARS-CoV-2 AY.44 | SARS-CoV-2 AY.45 | Mpox B.1 |
| 20 | 6 | 158 | 123 |
| 15 | 5 | 150 | 121 |
| 10 | 4 | 49 | 67 |
| 7 | 4 | 26 | 45 |
| 5 | 2 | 13 | 23 |
| 3 | 2 | 5 | – |
| 1 | 2 | 2 | – |

Number (and percentage, out of a total of 500) of sequences detected as 1BP/2BP recombinants or non-recombinants by RecombinHunt (RH). b Specificity analysis. Number (and percentage, out of a total of 500) of sequences detected as 1BP/2BP recombinants when they instead were non-recombinants. c Minimum number of sequences to correctly characterize a real lineage. Cell values represent the median value $\hat{N}$ such that the median number of characteristic mutations is correct for all the subsequent values of $N \geq \hat{N}$. The median values have been computed by sampling N sequences 100 times for increasing values of N. Each mutations-noise level indicates the maximum number of mutations that differ from the characterization of the lineage, present in the sequences of the sampled dataset.

Real-world scenarios can only be approximated by our stochastic model, which ignores linkage disequilibrium, sampling biases, convergent evolution and/or errors in sequencing and classification. To capture real-world variability, we considered a selection of random SARS-CoV-2 and monkeypox lineages[49]. For every sequence S assigned to a lineage L, we measured the *mutations-noise* as the number of mutations in S not included in the set of characteristic mutations of L; we then partitioned the dataset into discrete subsets with a level of mutations-noise less than certain discrete thresholds. We randomly sampled N sequences of L 100 times, while varying mutations-noise levels, and measured the minimum value of $N$, $\hat{N}$, such that for all the values of $N \geq \hat{N}$ the median number of characteristic mutations, as determined by the analysis of the sampled sequences, was identical to the number of L characteristic mutations. $\hat{N}$ indicates the minimum number of sequences that should be available, in SARS-CoV2 and monkeypox, to yield sufficient stability of the characteristic mutations. Relevant combinations of mutations-noise levels and $\hat{N}$ are reported in Table 1c.

**Lineage analysis using consensus-genomes**

Our method was executed on 51 of the 57 lineages designated as recombinant by Pango at the end of COVID-19 pandemic emergency (April 2023). Two lineages were excluded since they had three breakpoints and other four lineages were disregarded since the defining Pango issue was unclear/controversial. The "ground truth", i.e., the description of the recombinant lineage in terms of donor, acceptor, and breakpoints, was reconstructed directly from the corresponding Pango designation issues[43,50].

Recombinant lineages were removed from the list of Pango designations, and an ideal consensus-genome was reconstructed for every lineage by considering the ordered list of nucleotide mutations shared by >75% high-quality genomes assigned to that lineage.

Three main criteria were used to validate RecombinHunt results against the ground truth: (1) the correct model recombinant (1BP or 2BP) represented a statistically significant improvement (p-value $< 10^{-2}$), compared with the other models; (2) designations of both the donor and the acceptor lineages were correct (i.e., all found candidates are the same or descendants of those in the ground truth); and (3) the breakpoint position, as determined by our method, was within the range of genomic positions indicated in the Pango issue.

Results are shown in Table 2, partitioned as follows: (i) cases with 1BP approximately in the middle of the SARS-CoV-2 genome (from XA to XZ); (ii) 5' proximal 1BP cases (from XAA to XW); (iii) 3' proximal 1BP cases (from XAH to XT); (iv) cases with 2BP (from XAC to XD). Supplementary Notes 2 collect the visual analyses for all Pango recombination cases analyzed in GISAID data.

RecombinHunt results were in complete agreement with the ground truth for 40 recombinant lineages (37 with one breakpoint and 3 with two breakpoint recombinations). The remaining 11 lineages – which did not fully agree with the Pango designation – are stratified into three conceptually distinct groups: G1, G2, and G3; a detailed report of these cases is in Supplementary Notes 5.

(G1) Six lineages (XAV, XAR, XBF, XN, XAK, and XAZ) are not flagged as recombinant by RecombinHunt. For all these lineages, recombination is supported only by one mutation (XAK and XAZ) or two mutations (XAV, XAR, XBF, and XN), over an average of 67 mutations considered in the respective consensus-genome. Given the limited number of mutations supporting the recombination events, and the constraints used by RecombinHunt, these cases fall outside the scope of application of the method.

## Table 2 | Summary of results on GISAID dataset

| Lin | #seq | #mut | GT lineages | GT BP | RH lineage candidates | RH BP | RM, LC, BP |
|---|---|---|---|---|---|---|---|
| XA | 7 | 32 | B.1.177 + B.1.1.7 | 12–14 | B.1.177.18 + B.1.1.7 | 13–14 | ✓(3.12e-94), ✓, ✓ |
| XAD | 70 | 65 | BA.2* + BA.1* | 54–56 | BA.2 + BA.1.14.1 | 55–56 | ✓(7.19e-24), ✓, ✓ |
| XAE | 50 | 67 | BA.2* + BA.1* | 53–56 | BA.2 + BA.1.14 | 55–56 | ✓(3.06e-26), ✓, ✓ |
| XAL | 72 | 61 | BA.1* + BA.2* | 15–17 | BA.1.1 + BA.2 | 15–16 | ✓(6.09e-76), ✓, ✓, |
| XAN | 123 | 71 | BA.2* + BA.5.1 | 21–28 | BJ.1 + BA.5.1.23 | 11–12 | ✓(3.15e-16), ✓, ✗ |
| XAQ | 5 | 64 | BA.1* + BA.2* | 17–18 | BA.1 + BA.2 | 16–17 | ✓(3.26e-72), ✓, ✓ |
| XAV | 42 | 70 | BA.2* + BA.5* | 19–22 | BA.5.1.24 | Non-rec | —— ✗ (G1) —— |
| XBB | 1752 | 85 | BJ.1 + BM.1.1.1 | 49–52 | BJ.1 + BM.1.1.1 | 51–52 | ✓(2.99e-87), ✓, ✓ |
| XBD | 141 | 82 | BA.2.75.2 + BA.5.2.1 | 52–66 | BA.2.75.2 + BF.3 | 65–66 | ✓(4.78e-64), ✓, ✓ |
| XBE | 63 | 72 | BA.5.2* + BE.4.1 | 31–53 | BA.5.2.6 + BE.4 | 63–64 | ✓(5.22e-26), ✓, ✗ |
| XBF | 4944 | 84 | BA.5.2 + CJ.1 | 14–16 | BM.1.1.1 | Non-rec | —— ✗ (G1) —— |
| XBG | 86 | 75 | BA.2.76 + BA.5.2 | 32–41 | BA.2.76 + BA.5.2 | 40–41 | ✓(3.95e-50), ✓, ✓ |
| XBH | 71 | 78 | BA.2.3.17 + BA.2.75.2 | 19–28 | BA.2.75.2 + BA.2.3.17 + BA.2.75.2 | 4–5, 19–20 | —— ✗ (G2) —— |
| XBJ | 120 | 88 | BA.2.3.20 + BA.5.2* | 59–73 | BA.2.3.20 + BA.5.2.36 | 72–73 | ✓(8.54e-89), ✓, ✓ |
| XBM | 205 | 77 | BA.2.76 + BF.3 | 33–42 | BF.3 + BA.2.76 + BF.3 | 6–7, 33–34 | —— ✗ (G2) —— |
| XE | 1009 | 63 | BA.1* + BA.2* | 9–11 | BA.1.17.2 + BA.2.29 | 9–10 | ✓(2.70e-106), ✓, ✓ |
| XJ | 47 | 60 | BA.1* + BA.2* | 13–16 | BA.1.17.2 + BA.2.65 | 13–14 | ✓(2.59e-123), ✓, ✓ |
| XK | 25 | 55 | BA.1* + BA.2* | 15–17 | BA.1.7 + BA.2.13 | 15–16 | ✓(1.99e-185), ✓, ✓ |
| XM | 301 | 59 | BA.1.1* + BA.2* | 14–17 | BA.1.1.16 + BA.2.33 | 14–15 | ✓(2.17e-217), ✓, ✓ |
| XV | 27 | 61 | BA.1* + BA.2* | 12–14 | BA.1.6 + BA.2.25 | 12–13 | ✓(1.73e-167), ✓, ✓ |
| XY | 44 | 68 | BA.1* + BA.2* | 14–16 | BA.1.1 + BA.2 | 14–15 | ✓(3.43e-67), ✓, ✓ |
| XZ | 49 | 65 | BA.2* + BA.1* | 55–56 | BA.2 + BA.1.1.12 | 56–57 | ✓(2.84e-23), ✓, ✓ |
| XAA | 33 | 67 | BA.1* + BA.2* | 7–8 | BA.1.20 + BA.2 | 7–8 | ✓(3.56e-37), ✓, ✓ |
| XAB | 73 | 67 | BA.1* + BA.2* | 6–7 | BA.1.1.16 + BA.2 | 6–7 | ✓(2.86e-48), ✓, ✓ |
| XAF | 64 | 65 | BA.1* + BA.2* | 8–10 | BA.1.1.16 + BA.2.9 | 9–10 | ✓(8.17e-116), ✓, ✓ |
| XAG | 227 | 70 | BA.1* + BA.2* | 8–9 | BA.1.1.14 + BA.2 | 6–7 | ✓(1.33e-36), ✓, ✗ |
| XAM | 147 | 69 | BA.1.1 + BA.2.9 | 6–7 | BA.1.1.16 + BA.2.9 | 6–7 | ✓(4.72e-55), ✓, ✓ |
| XAR | 12 | 64 | BA.1* + BA.2* | 1–3 | BA.2 | Non-rec | —— ✗ (G1) —— |
| XAU | 72 | 69 | BA.1.1* + BA.2.9* | 3–5 | BA.1.1.2 + BA.2.9 | 3–4 | ✓(2.51e-25), ✓, ✓ |
| XF | 12 | 62 | B.1.617.2* + BA.1* | 6–7 | AY.37 + BA.1.16 | 6–7 | ✓(2.96e-21), ✓, ✓ |
| XG | 272 | 66 | BA.1* + BA.2* | 6–7 | BA.1.17 + BA.2 | 6–7 | ✓(2.82e-29), ✓, ✓ |
| XH | 93 | 65 | BA.1* + BA.2* | 10–12 | BA.1 + BA.2.9 | 10–11 | ✓(5.96e-64), ✓, ✓ |
| XL | 28 | 69 | BA.1* + BA.2* | 8–9 | BA.1.17.2 + BA.2 | 8–9 | ✓(1.36e-42), ✓, ✓ |
| XN | 82 | 66 | BA.1* + BA.2* | 2–4 | BA.2 | Non-rec | —— ✗ (G1) —— |
| XQ | 43 | 64 | BA.1.1* + BA.2* | 4–5 | BA.1.1.16 + BA.2.5 | 4–5 | ✓(3.77e-61), ✓, ✓ |
| XR | 91 | 66 | BA.1.1* + BA.2* | 4–5 | BA.1.1.16 + BA.2 | 4–5 | ✓(8.24e-42), ✓, ✓ |
| XS | 15 | 64 | B.1.617.2* + BA.1.1* | 10–12 | AY.126 + BA.1.1 | 10–11 | ✓(8.23e-63), ✓, ✓ |
| XU | 4 | 63 | BA.1* + BA.2* | 5–6 | BA.1.17 + BA.2.37 | 5–6 | ✓(8.39e-51), ✓, ✓ |
| XW | 88 | 67 | BA.1* + BA.2* | 3–5 | BA.1.1.2 + BA.2.23 | 3–4 | ✓(4.44e-35), ✓, ✓ |
| XAH | 141 | 64 | BA.2* + BA.1* | 59–61 | BA.2 + BA.1.17 | 56–57 | ✓(2.49e-13), ✓, ✗ |
| XAP | 26 | 64 | BA.2* + BA.1* | 54–56 | BA.2 + BA.1.1.12 | 55–56 | ✓(2.84e-23), ✓, ✓ |
| XAT | 28 | 66 | BA.2.3.13 + BA.1* | 57–59 | BA.2.3.13 | Non-rec | —— ✗ (G3) —— |
| XC | 4 | 36 | AY.29 + B.1.1.7 | 27–28 | AY.29.1 + Q.1 | 28–29 | ✓(3.45e-98), ✓, ✓ |
| XP | 12 | 66 | BA.1.1* + BA.2* | 58–65 | BA.1.1 | Non-rec | —— ✗ (G3) —— |
| XT | 11 | 61 | BA.2* + BA.1* | 52–54 | BA.2 + BA.1.22 | 53–54 | ✓(1.67e-29), ✓, ✓ |
| XAC | 33 | 68 | BA.2* + BA.1* + BA.2* | 56–58, 62–68 | BA.2.3 + BA.1.1.16 + BA.2.3 | 56–57, 62–63 | ✓(1.38e-19), ✓, ✓ |
| XAK | 110 | 69 | BA.2* + BA.1* + BA.2* | 20–23, 23–24 | BA.2 | Non-rec | —— ✗ (G1) —— |
| XAW | 28 | 103 | AY.122 + BA.2* + AY.122 | 48–53, 91–93 | AY.122 + BQ.1.12 + AY.122 | 51–52, 97–98 | —— ✗ (G3) —— |
| XAZ | 1390 | 69 | BA.2.5 + BA.5 + BA.2.5 | 8–14, 63–64 | BQ.1.9 + BA.5 | 3–4 | —— ✗ (G1) —— |
| XBL | 153 | 92 | XBB.1 + BA.2.75 + XBB.1 | 2–7, 12–22 | XBB.1.5 + BN.1.3 + XBB.1.5 | 5–6, 12–13 | ✓(8.60e-16), ✓, ✓ |
| XD | 14 | 67 | B.1.617.2* + BA.1* + B.1.617.2* | 23–26, 51–53 | AY.4 + BA.1.22 + AY.4 | 23–24, 52–53 | ✓(2.21e-170), ✓, ✓ |

Four horizontal sections represent the cases that in the Pango lineage designation ground truth (GT) are defined as i) 1BP, with a breakpoint in the central part of the genome; ii) 1BP, with a breakpoint close to the genome 5′-end; iii) 1BP (with a breakpoint close to the genome 3′-end; and iv) 2BP. The table columns represent, respectively: name of Pango lineage; number of high-quality sequences assigned to the lineage in the database; number of mutations in the 75% consensus-genome for the observed lineage, defining the mutations-space; lineages in the GT; GT breakpoint coordinates in the mutations-space; candidates found by RecombinHunt (RH); RH breakpoint coordinates in the mutations-space; comparison between GT and RH results. Here, three checks are reported. (RM) Recombination Model: a checkmark when RH selected the same model as GT (non-recombinant, 1BP or 2BP), a crossmark otherwise–with the *p*-value of the one-sided AIC comparison -without multiple comparison correction- between the recombination model and the non-recombination one; (LC) Lineage Candidates: a checkmark when all RH candidates are the same or descendants of those in GT, a crossmark otherwise; (BP) BreakPoint Range: a checkmark when the BP coordinates in RH are included in those of GT with at most 1 mutation of difference, a crossmark otherwise. In eleven cases the output of RecombinHunt is considerably different from the GT. These are indicated with a crossmark and a discussion code G1–G3, detailed in the text.

**Table 3 | Summary of single-sequence analysis results on 100 (or less) available genomes for each recombinant lineage**

| Lin | #seq | %non-rec | %1BP | %2BP | %p < e-5 | Consensus | |
|-----|------|----------|------|------|----------|-----------|---|
| XA | 7 | 0 | **1** | 0 | 1 | 1BP | |
| XAD | 70 | 0.13 | **0.71** | 0.16 | 0.97 | 1BP | |
| XAE | 50 | 0.02 | **0.98** | 0 | 1 | 1BP | |
| XAL | 72 | 0 | **1** | 0 | 1 | 1BP | |
| XAN | 100 | 0.28 | **0.64** | 0.08 | 0.98 | 1BP | |
| XAQ | 5 | 0 | **1** | 0 | 1 | 1BP | |
| XAV | 42 | **0.62** | 0.26 | 0.12 | 0.98 | Non-rec | |
| XBB | 100 | 0 | **0.92** | 0.08 | 0.99 | 1BP | |
| XBD | 100 | 0 | **0.8** | 0.2 | 1 | 1BP | |
| XBE | 63 | 0 | **0.95** | 0.05 | 1 | 1BP | |
| XBF | 100 | **0.91** | 0.05 | 0.04 | 0.99 | Non-rec | |
| XBG | 86 | 0.01 | 0.3 | <u>0.69</u> | 1 | 1BP | * |
| XBH | 71 | 0 | <u>0.86</u> | 0.14 | 1 | 2BP | * |
| XBJ | 100 | 0 | **1** | 0 | 1 | 1BP | |
| XBM | 100 | 0 | <u>0.56</u> | 0.44 | 1 | 2BP | * |
| XE | 100 | 0.01 | **0.99** | 0 | 0.99 | 1BP | |
| XJ | 47 | 0.02 | **0.98** | 0 | 0.98 | 1BP | |
| XK | 25 | 0 | **1** | 0 | 1 | 1BP | |
| XM | 100 | 0 | **1** | 0 | 1 | 1BP | |
| XV | 27 | 0 | **1** | 0 | 1 | 1BP | |
| XY | 44 | 0 | **1** | 0 | 1 | 1BP | |
| XZ | 49 | 0 | **0.98** | 0.02 | 1 | 1BP | |
| XAA | 33 | 0 | **1** | 0 | 1 | 1BP | |
| XAB | 73 | 0.01 | **0.99** | 0 | 1 | 1BP | |
| XAF | 64 | 0 | **1** | 0 | 1 | 1BP | |
| XAG | 100 | 0 | **1** | 0 | 1 | 1BP | |
| XAM | 100 | 0 | **1** | 0 | 1 | 1BP | |
| XAR | 12 | **1** | 0 | 0 | 1 | Non-rec | |
| XAU | 72 | 0.03 | **0.97** | 0 | 1 | 1BP | |
| XF | 12 | 0.08 | **0.92** | 0 | 1 | 1BP | |
| XG | 100 | 0 | **1** | 0 | 1 | 1BP | |
| XH | 93 | 0 | **0.99** | 0.01 | 1 | 1BP | |
| XL | 28 | 0 | **1** | 0 | 1 | 1BP | |
| XN | 82 | **0.98** | 0.02 | 0 | 0.99 | Non-rec | |
| XQ | 43 | 0 | **1** | 0 | 1 | 1BP | |
| XR | 91 | 0.18 | **0.82** | 0 | 1 | 1BP | |
| XS | 15 | 0 | **1** | 0 | 1 | 1BP | |
| XU | 4 | 0 | **1** | 0 | 1 | 1BP | |
| XW | 88 | 0.11 | **0.89** | 0 | 0.99 | 1BP | |
| XAH | 100 | 0.05 | **0.95** | 0 | 0.98 | 1BP | |
| XAP | 26 | 0.19 | **0.81** | 0 | 1 | 1BP | |
| XAT | 28 | 0.43 | <u>0.57</u> | 0 | 0.96 | Non-rec | * |
| XC | 4 | 0 | **1** | 0 | 1 | 1BP | |
| XP | 12 | **0.92** | 0.08 | 0 | 1 | Non-rec | |
| XT | 11 | 0 | **1** | 0 | 1 | 1BP | |
| XAC | 33 | 0.15 | 0.21 | **0.64** | 0.85 | 2BP | |
| XAK | 100 | **0.71** | 0.13 | 0.16 | 0.06 | Non-rec | |
| XAW | 28 | 0.04 | 0.18 | **0.79** | 0.93 | 2BP | |
| XAZ | 100 | 0.13 | <u>0.47</u> | 0.4 | 0.83 | 1BP | * |
| XBL | 100 | 0.02 | 0.03 | **0.95** | 1 | 2BP | |
| XD | 14 | 0 | 0 | **1** | 1 | 2BP | |

For each Pango lineage, we indicate the number of considered high-quality sequences—100 random ones are selected when more are available. In the following columns, we indicate percentages of sequences with no breakpoint, one breakpoint, or two breakpoints (bold-type for solid majority). The following column reports the percentage of sequences for which the p-value (of one-sided AIC comparison – without multiple comparison correction), that establishes the most probable model for each sequence, is ≤10⁻⁵. The last column reports the model chosen for the consensus-genome; an asterisk indicates when that model is in contrast with the majority of sequences (underlined).

(G2) In two cases (XBH and XBM), RecombinHunt identifies the same parent lineages, but additional breakpoints (2BP wrt 1BP) compared with the solutions reported by the ground truth. These results might indicate that our approach can deconvolute complex patterns of recombination that could not be easily inferred by manual analyses.

(G3) Three cases (XAW, XAT, and XP) highlight some limitations of our approach. In XAW, while L1 is correctly identified, none of the lineages defined in the Pango nomenclature provides a good match for 46 out of 103 total target mutations characteristic of the XAW lineage; in this case, the parent acceptor lineage might not be defined in the reference nomenclature. In XAT and XP, recombination as defined by the ground truth is supported by a limited number of mutations at the 3′-end of the genome. As these mutations have a relatively high frequency (range 0.17–0.34) in the complete collection of SARS-CoV-2 genomes, they contribute to a modest drop in the log-likelihood ratio score. Both XAT and XP lineages are flagged as non-recombinant by RecombinHunt; we speculate that our method loses sensitivity when recombination events are supported only by a limited number of mutations, localized at the terminal ends of the genome and associated with a relatively high frequency in the viral population. However, under the above scenario, where mutations that support a recombination event are few and occur in the viral population with relatively high frequencies, convergent evolution, and positive selection could represent an equally plausible alternative model to recombination.

## Analysis of single high-quality sequences

To evaluate whether RecombinHunt could systematically flag recombinant viral genomes without assuming a prior assignment to a recombinant lineage, our method was applied directly to single genome sequences. For every recombinant lineage, at most one hundred randomly selected, high-quality sequences were analyzed; all the available sequences were considered for lineages with less than 100 high-quality sequences assigned, instead.

Results are shown in Table 3 (in the same order and grouping of Table 2, to facilitate the comparison). In 46 out of 51 lineages, RecombinHunt's results for the large majority of single genome sequences (value in bold type, either '%non-rec', '%1BP', or '%2BP') are consistent with those obtained on the corresponding lineage-level consensus-genome. The most notable exceptions are: XBG/XAZ (consensus 1BP, whereas respectively 69% and 40% of sequences are 2BP); XBH/XBM (consensus 2BP, whereas respectively 86% and 56% of sequences are 1BP); and XAT (consensus is non-recombinant, whereas 57% of sequences are 1BP).

Histograms in Fig. 4 report the observed frequencies of inferred breakpoint genomic positions. In the first three groups of plots (from XA to XT), for 29 lineages the distribution of results obtained on single sequences is in large agreement with the analysis at the consensus-genome level – see that the mode of the blue bar plots is close to the light blue bar, indicating the breakpoint position of the consensus-genome. Some discrepancies however are observed, which can be summarized in two main cases. (1) Additional breakpoints. The consensus-genome analysis indicated one breakpoint, but two breakpoints were detected in most individual sequences. This occurs in eight lineages. XBG is the most evident case: the corresponding panel in Fig. 4 shows a single light-blue bar (breakpoint based on consensus) and a prevalence of single sequences with 2 predicted breakpoints (two orange bars). (2) Missing breakpoints. Two breakpoints were identified based on the consensus-genome analysis, but only one breakpoint is detected in single sequences: this occurs in two lineages XBH and XBM, whose panels in Fig. 4 show two light orange bars and several blue bars, representing single sequences with one breakpoint.

Note that in XAC, XAW, XBL, and XD both the consensus-genome and the majority of sequences were recognized as two-breakpoint recombinants, and that all single genome sequences assigned to XAR

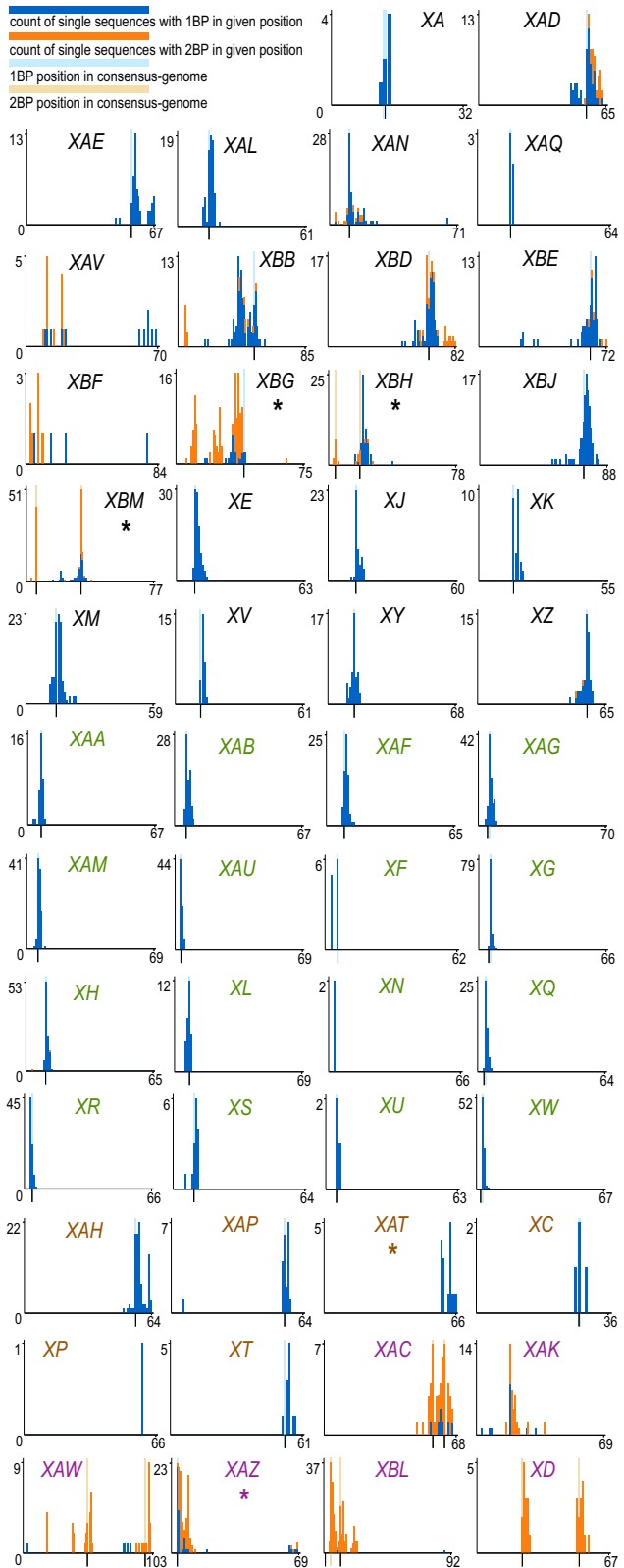

**Fig. 4 | Barplots of RecombinHunt (RH) outputs for breakpoints positions.**
Label colors reflect the four groups in Table 3. On the x-axis, the consensus-genome mutations; on the y-axis, the count of single sequences with a breakpoint detected on the x position. A light blue stripe indicates the RH 1BP position; two light orange stripes indicate the RH 2BP positions (see Table 2). Blue bar plots count the sequences whose 1BP is located at a given mutation; orange bar plots count the sequences whose 2BPs are located at given positions (without distinction between the first and the second one).

were flagged as non-recombinant, in agreement with the consensus-genome analysis result.

## Post hoc detection of recombinant lineages

The dataset analyzed in this manuscript represents a data freeze of the GISAID database as of April 1st, 2023. At that time, the recombinant lineages XCA and XCB had not yet been designated by the Pango community. These lineages were first introduced in Pango, respectively, on April 3rd, 2023 (XCA: BA.2.75 + BQ.1)[51] and on April 14th, 2023 (XCB: BF.31.1 + BQ.1.10)[52]. Since our dataset already included 13 XCA-projected sequences (whose designation was subsequently changed to XCA) and 11 XCB-projected sequences (subsequently changed to XCB), we executed post-hoc analyses on these two groups of sequences to evaluate the predictive power of RecombinHunt and its potential application for the early identification of recombinant sequences.

All of the 13 XCA-projected sequences were correctly labeled as recombinants of BA.2.75 and BQ.1 (12:1BP, 1:2BP) by RecombinHunt. Also the 11 XCB-projected sequences were all recognized as recombinants, in the majority of cases (8/11) with the parent lineages corresponded to the same designation indicated by Pango (i.e., as recombinants of BF.31.1 and BQ.1.10). In the remaining 3 cases, the method returned slightly different combinations of parent lineages, still compatible with the Pango designation of XCB.

These results show that both XCA-projected and XCB-projected sequences would have been correctly classified as recombinant by RecombinHunt, and suggest that our method could contribute a significant advance in the early identification of candidate recombinant sequences/lineages.

## Comparison with the RIPPLES method

Turakhia et al.[6] introduced RIPPLES, an elegant method based on parsimony analyses for the detection of recombination in SARS-CoV-2. Briefly, candidate recombinant sequences are partitioned into discrete genomic segments. Subsequently, each segment is placed on the SARS-CoV-2 global phylogeny by maximum parsimony; inconsistent phylogenetic signals across genome segments are detected and reconciled; finally, the donor and acceptor lineages are identified as the lineages that result in the highest parsimony score improvement relative to the original placement on the global phylogenetic tree.

The same authors developed RIVET[44] – an extension of RIPPLES that allows the systematic monitoring of SARS-CoV-2 genome sequences, and the flagging of recombinant genomes in real-time. At the time of writing RIVET is probably the most complete method for the detection of novel recombinants in SARS-CoV-2. The authors claimed that "RIVET inferences (such as lineages of parent sequences) of known recombinants were largely consistent with those of manual curators", however, a systematic assessment was not provided. We accessed RIVET results as based on the public March 30th 2023 release of the Nextstrain[53] curated collection of SARS-CoV-2 genome sequences[54], and performed a comparison with the results obtained by RecombinHunt on the same dataset. Since RIVET results were retrieved directly from its public endpoint we assumed them to reflect the optimal parameter configuration for this method.

The Nextstrain dataset was processed according to our quality criteria (see Methods). Out of a total of 6,983,419 sequences, 3,984,308 sequences, and 61 distinct recombinant Pango lineages were retained. After removing the cases with three or more breakpoints and those for which the ground truth is uncertain, we obtained 51 recombinant cases. A consensus set of characteristic mutations was computed for these lineages by retaining mutations with a frequency above the 75% threshold. Detailed results of RecombinHunt are reported in Table 4; Supplementary Notes 3 collect the visual analyses for all Pango recombination cases analyzed in Nextstrain data.

RecombinHunt labels 43 cases correctly w.r.t. the ground truth; these include two lineages (XBB, XM) reporting a different (but

**Table 4 | Summary of results on Nextstrain dataset**

| Lin | #seq | #mut | GT lineages | GT BP | RH lineage candidates | RH BP | RM, LC, BP |
|-----|------|------|-------------|-------|------------------------|-------|------------|
| XA | 33 | 36 | B.1.177 + B.1.1.7 | 12–14 | B.1.177.18 + B.1.1.7 | 13–14 | ✓(6.86e-101), ✓, ✓ |
| XAD | 11 | 69 | BA.2* + BA.1* | 55–57 | BA.2 + BA.1 | 56–57 | ✓(4.11e-21), ✓, ✓ |
| XAE | 18 | 72 | BA.2* + BA.1* | 55–58 | BA.2 + BA.1 | 57–58 | ✓(1.11e-28), ✓, ✓ |
| XAL | 13 | 67 | BA.1* + BA.2* | 17–19 | BA.1.1 + BA.2 | 17–18 | ✓(1.96e-86), ✓, ✓ |
| XAN | 36 | 72 | BA.2* + BA.5.1 | 21–28 | BA.2 + BA.5.1.23 | 11–12 | ✓(4.03e-17), ✓, ✗ |
| XAP | 261 | 65 | BA.2* + BA.1* | 53–55 | BA.2.65 + BA.1.23 | 54–55 | ✓(9.79e-70), ✓, ✓ |
| XAT | 3 | 66 | BA.2.3.13 + BA.1* | 55–57 | BA.2.3 | Non-rec | —— ✗(G3) —— |
| XAV | 19 | 72 | BA.2* + BA.5* | 19–22 | BA.5.1.24 | Non-rec | —— ✗(G1) —— |
| XBB | 162 | 92 | BJ.1 + BM.1.1.1 | 50–54 | BA.2.9 + BM.1.1.1 | 38–39 | ✓(1.25e-66), ✓*, ✗ |
| XBD | 66 | 86 | BA.2.75.2 + BA.5.2.1 | 53–67 | BA.2.75.2 + BF.3 | 66–67 | ✓(2.05e-85), ✓, ✗ |
| XBE | 134 | 75 | BA.5.2* + BE.4.1 | 31–53 | BA.5.2.6 + BE.4.1.1 | 63–64 | ✓(2.32e-38), ✓, ✗ |
| XBF | 298 | 91 | BA.5.2 + CJ.1 | 14–16 | BA.5.2.3 + CJ.1.1 | 11–12 | ✓(1.27e-56), ✓, ✗ |
| XBG | 36 | 78 | BA.2.76 + BA.5.2 | 32–41 | BA.2.76 + BA.5.2 | 23–24 | ✓(8.88e-44), ✓, ✗ |
| XBH | 10 | 84 | BA.2.3.17 + BA.2.75.2 | 19–28 | BA.2.75.2 + BA.2.3.17 + BA.2.75.2 | 4–5, 19–20 | —— ✗(G2) —— |
| XBJ | 9 | 92 | BA.2.3.20 + BA.5.2* | 60–74 | BA.2.3.20 + BA.5.2.62 | 73–74 | ✓(8.73e-113), ✓, ✓ |
| XBM | 49 | 77 | BA.2.76 + BF.3 | 31–40 | BA.2.76 + BF.3.1 | 31–32 | ✓(1.40e-69), ✓, ✓ |
| XBP | 23 | 88 | BA.2.75* + BQ.1* | 40–42 | BL.1 + BQ.1.1.3 | 41–42 | ✓(1.04e-150), ✓, ✓ |
| XBR | 8 | 94 | BA.2.75 + BQ.1 | 41–42 | BN.3.1 + BQ.1.25.1 | 41–42 | ✓(4.73e-205), ✓, ✓ |
| XBW | 5 | 95 | XBB.1.5 + BQ.1.14 | 76–80 | XBB.1.5 + BQ.1.14 | 79–80 | ✓(9.08e-72), ✓, ✓ |
| XJ | 24 | 65 | BA.1* + BA.2* | 11–14 | BA.1.24 + BA.2.1 | 11–12 | ✓(1.22e-182), ✓, ✓ |
| XM | 145 | 62 | BA.1.1* + BA.2* | 13–16 | BA.1.24 + BA.2.27 | 13–14 | ✓(4.56e-200), ✓*, ✗ |
| XV | 25 | 66 | BA.1* + BA.2* | 12–14 | BA.1.14 + BA.2.52 | 12–13 | ✓(5.14e-165), ✓, ✓ |
| XY | 62 | 74 | BA.1* + BA.2* | 14–16 | BA.1.1 + BA.2 | 14–15 | ✓(8.66e-77), ✓, ✓ |
| XZ | 140 | 66 | BA.2* + BA.1* | 54–55 | BA.2.34 + BA.1.23 | 55–56 | ✓(8.68e-70), ✓, ✓ |
| XAA | 52 | 73 | BA.1* + BA.2* | 7–8 | BA.1.20 + BA.2 | 7–8 | ✓(9.03e-37), ✓, ✓ |
| XAB | 88 | 67 | BA.1* + BA.2* | 4–5 | BA.1.6 + BA.2.27 | 4–5 | ✓(4.30e-61), ✓, ✓ |
| XAF | 51 | 66 | BA.1* + BA.2* | 8–10 | BA.1.1.9 + BA.2.7 | 8–9 | ✓(1.41e-130), ✓, ✓ |
| XAG | 15 | 74 | BA.1* + BA.2* | 8–9 | BA.1.1.14 + BA.2.9 | 8–9 | ✓(5.25e-38), ✓, ✓ |
| XAM | 67 | 74 | BA.1.1 + BA.2.9 | 6–7 | BA.1.1.9 + BA.2.9 | 6–7 | ✓(7.24e-52), ✓, ✓ |
| XAR | 65 | 69 | BA.1* + BA.2* | 2–4 | BA.2.23 | Non-rec | —— ✗(G1) —— |
| XAU | 17 | 71 | BA.1.1* + BA.2.9* | 3–5 | BA.1.1.9 + BA.2.9 | 4–5 | ✓(1.08e-25), ✓, ✓ |
| XE | 1342 | 67 | BA.1* + BA.2* | 9–11 | BD.1 + BA.2.31 | 10–11 | ✓(6.83e-134), ✓, ✓ |
| XF | 19 | 64 | B.1.617.2* + BA.1* | 6–7 | AY.4.2 + BA.1.16 | 6–7 | ✓(9.20e-46), ✓, ✓ |
| XG | 305 | 70 | BA.1* + BA.2* | 6–7 | BA.1.17 + BA.2 | 6–7 | ✓(4.42e-49), ✓, ✓ |
| XH | 102 | 68 | BA.1* + BA.2* | 10–12 | BA.1 + BA.2.9 | 10–11 | ✓(6.81e-61), ✓, ✓ |
| XL | 70 | 74 | BA.1* + BA.2* | 8–9 | BA.1.17.2 + BA.2 | 8–9 | ✓(1.83e-39), ✓, ✓ |
| XN | 167 | 71 | BA.1* + BA.2* | 2–4 | BA.2 | Non-rec | —— ✗(G1) —— |
| XQ | 90 | 68 | BA.1.1* + BA.2* | 4–5 | BA.1.1.9 + BA.2.23 | 4–5 | ✓(1.32e-46), ✓, ✓ |
| XR | 24 | 72 | BA.1.1* + BA.2* | 4–5 | BA.1.1.15 + BA.2.9 | 5–6 | ✓(2.23e-32), ✓, ✓ |
| XS | 24 | 68 | B.1.617.2* + BA.1.1* | 10–12 | AY.126 + BA.1.1 | 10–11 | ✓(3.01e-68), ✓, ✓ |
| XU | 6 | 71 | BA.1* + BA.2* | 6–7 | BA.1.10 + BA.2 | 6–7 | ✓(4.38e-47), ✓, ✓ |
| XW | 47 | 72 | BA.1* + BA.2* | 3–5 | BA.1.1.9 + BA.2 | 3–4 | ✓(1.85e-25), ✓, ✓ |
| XAH | 1 | 71 | BA.2* + BA.1* | 63–65 | BA.2 + BA.1.1.18 | 58–59 | ✓(1.33e-22), ✓*, ✗ |
| XP | 56 | 68 | BA.1.1* + BA.2* | 58–67 | BA.1.1 + BA.5.6.3 | 59–60 | —— ✗(G3) —— |
| XAC | 34 | 71 | BA.2* + BA.1* + BA.2* | 56–58, 62–70 | BA.2.3 + BA.1.1.9 + BA.2.3 | 57–58, 62–63 | ✓(5.84e-46), ✓, ✓ |
| XAK | 5 | 73 | BA.2* + BA.1* + BA.2* | 20–23, 23–24 | BA.2 + BA.2.10 | Non-rec | —— ✗(G1) —— |
| XAZ | 289 | 71 | BA.2.5 + BA.5 + BA.2.5 | 8–14, 63–64 | BA.5.1.27 + BA.5 | Non-rec | —— ✗(G1) —— |
| XBL | 5 | 101 | XBB.1 + BA.2.75 + XBB.1 | 2–7, 12–22 | XBB.1.5.24 + BN.1.3 + XBB.1.5.24 | 3–4, 12–13 | ✓(1.54e-44), ✓, ✓ |
| XBT | 8 | 86 | BA.5.2.34 + BA.2.75 + BA.5.2.34 | 10–14, 38–50 | BA.5.2.34 + BL.1.5 + BA.5.2.34 | 13–14, 39–40 | ✓(1.21e-110), ✓, ✓ |
| XBU | 5 | 89 | BA.2.75* + BQ.1* + BA.2.75* | 40–53, 70–75 | BA.2.75.2 + BQ.1.1.19 + BA.2.75.2 | 51–52, 70–71 | ✓(7.62e-72), ✓, ✓ |
| XD | 10 | 69 | B.1.617.2* + BA.1* + B.1.617.2* | 25–27, 53–55 | AY.4 + BA.1.15.3 + AY.4 | 25–26, 54–55 | ✓(1.93e-166), ✓, ✓ |

Four horizontal sections and column definitions follow the same criteria as in Table 2. In two cases, XBB and XM, the last column contains a checkmark with asterisk when the lineage candidates (LC) correspond to the ancestor lineage of those in the ground truth.

phylogenetically close) ancestor lineage. A total of eight recombinant lineages (discussed in Supplementary Notes 6) are not flagged correctly by RecombinHunt; these fit within the categorization in the G1–G3 classes discussed above for the GISAID dataset, yielding to: (G1): XAV, XAR, XN, XAK, XAZ. (G2): XBH. (G3): XAT and XP.

RIVET reported a total of 17 inter-lineage distinct recombination cases on the complete Nextstrain dataset; out of these, two have more than two breakpoints and can not be addressed by RecombinHunt; moreover, XBN was not considered due to the uncertainty of the ground truth. Table 5a highlights that, out of 51 recombinant lineages recognized in Pango, RecombinHunt correctly identifies 43 cases. RIVET recognizes only 7 of them reporting parent lineages that are also Pango lineages and other 7 with donor/acceptor lineages that are not defined in Pango (e.g., miscBA.5BA.2.75 or miscBA1BA2Post17k).

Only 14 cases are recorded by both methods and allow a direct comparison (see Table 5b). For every lineage, the donor/acceptor candidates and breakpoints positions found by RecombinHunt and RIVET are compared against the ground truth. Note that in the seven cases (XAK, XBB, XBD, XD, XW, XBT, and XBU) RIVET reported donor/acceptor lineages not defined in Pango, thus candidate parent lineages could not be evaluated. Importantly, RecombinHunt reports 33 cases designated as recombinant in Pango that are not identified as recombinant by RIVET.

### RecombinHunt on monkeypox

The complete collection of viral genome sequences from the recent monkeypox epidemic[49] has 5402 records. Recombination has been reported in monkeypox; by applying a sophisticated approach based on the study of tandem repetitive sequences, Yeh et al.[55] identified eight distinct recombinant isolates, defined by an unusual arrangement of tandem repeat elements. Here, one had one breakpoint (Italy/FVG-ITA_01_2022), six had one or two breakpoints (ON838178.1, ON609725.2, ON754985.1, ON754986.1, ON754987.1, ON631241.1), and one had two or three breakpoints (ON631963.1). The authors also proposed an ad-hoc classification system, which however was not adopted by the scientific community. The currently WHO-accepted nomenclature for monkeypox[56] includes a total of 26 distinct lineages with no direct correspondence/equivalence with the custom designations defined by Yeh et al.

Monkeypox genome sequences and associated metadata were accessed through the Nextstrain resource[57], low-quality genome sequences were discarded according to the same criteria used for SARS-CoV-2, and mutations were identified by applying the HaploCoV workflow[27] on a collection of 2526 high-quality sequences. A total of 4932 distinct mutations were detected. Mutations characteristic of each lineage – as defined in the reference nomenclature – were identified. A threshold of 9% was applied, to account for the high levels of intra-lineage diversity observed in monkeypox[58].

A total of 374 1BP and 331 2BP candidate recombinant genomes were identified by RecombinHunt (see Supplementary Data 2). These include five of the eight manually flagged sequences by Yeh et al. (see Table 5c for details). Sequences ON838178.1, ON609725.2, ON754985.1, ON754986.1, and ON754987.1 – originally designated as lineage B.1.3 – were labeled as 2BP recombinants (B.1.3 + B.1.2 + B.1.3) by RecombinHunt. The 1st breakpoint was set at position 31-32 in the target mutations-space (~85k in the genome) for all the sequences, whereas the 2nd breakpoint was set at position 49-50 (~150k) for ON609725.2 and ON754987.1 and at 55-56 (~174k) for ON754986.1, ON754985.1, ON838178.1 cases. The recombination pattern inferred by RecombinHunt was highly consistent with the description in Yeh et al.[55], e.g., a 2BP recombination, U + M + U, in the custom nomenclature system defined by the authors. Note that the three sequences flagged as recombinant by Yeh et al. – but identified as non-recombinant by RecombinHunt – were assigned to the B.1 lineage in the reference monkeypox nomenclature system.

## Discussion

We introduce RecombinHunt, a purely data-driven method, for the identification of recombinant viral genomes in epidemic/pandemic scenarios. We record the frequency of nucleotide mutations occurring within lineages, as defined by a reference nomenclature (or otherwise determined clusters), and within all the genomes of a given sequence collection. These data are subsequently used to score viral genomes by a likelihood-based approach and detect recombinant sequences of two lineages, respectively labeled as the "donor" and the "acceptor". RecombinHunt is highly computationally efficient, and can be applied to the analysis of pandemic-scale data; the evaluation of the SARS-CoV-2 recombinant cases takes about 13 minutes on the GISAID dataset (15M sequences), and 8 minutes on the Nextstrain dataset (6.4M sequences) using a laptop.

The method is general and can be applied to many collections of viral genomes, as demonstrated by its application to SARS-CoV-2 (retrieved from the EpiCov database of GISAID and the GenBank database curated by Nextstrain) and monkeypox (Nextstrain). We observe that several viral pathogens, for which curated collections of genome sequences are available within Nextstrain and for which a structured nomenclature has been defined by the respective reference community, meet the minimum theoretical requirements for the application of RecombinHunt, with a sufficiently low mutations-noise level (e.g., below 10, see Table 1c). These include, for example, dengue, RSV, influenza, Enterovirus D68, and West Nile virus. Since the very high levels of sensitivity and specificity – when detecting true recombinations and avoiding false recombinations – our method could be confidently applied to any of these viruses, even in the absence of an established ground truth. Our minimum requirements analysis shows that the method is more accurate when used on large datasets, where classes are represented by a well-defined set of sequences and are well-separated from each other in the mutations-space. However, this does not prevent the application of RecombinHunt also to smaller datasets, with coarse-grained classification (see monkeypox).

RecombinHunt introduces a big data-oriented approach in the framework of likelihood-based methods for the detection of recombinant/mosaic genome structures. Two highly intertwined features define the key innovations of our method 1) usage of non-overlapping, potentially independent clusters of genome sequences, each characterized only by its prevalent mutations, for expressing the salient features of viral evolution; 2) a statistical framework built on the assessment of cumulative likelihood of a collection of characteristic mutations. RecombinHunt does not directly depend on phylogenetic inference; however, the need for the definition of a structured nomenclature and/or a discrete set of designations with coherent features represents a fundamental prerequisite for the application of RecombinHunt. At the time being, most systems for the nomenclature/classification of human pathogens are based on phylogenies and, in this context, having an accurate phylogeny is critical for determining the correct frequency threshold for the identification of characterizing mutations of distinct groups. The ideal value of this parameter depends, for a given viral species, on the size of the sequence datasets and on the granularity of the employed classification, and hence might be different for different use cases, as discussed in the specificity section.

Further, we recognize that RecombinHunt also presents some methodological limitations. First, it cannot detect recombination events supported by less than three mutations. Second, as we employ relative coordinates in the mutation-space rather than genomic coordinates (see also the gap-resolution procedure) the position of breakpoints is calculated with some degree of uncertainty. Nevertheless, none of these issues substantially impacts the ability of RecombinHunt to reliably construct a global SARS-CoV-2 recombination landscape.

**Table 5 | a RecombinHunt/RIVET Summary comparison**

**(a)**

| | Pango cases In Nextstrain | RH | RIVET Pango candidates | RIVET Non-Pango candidates |
|---|---|---|---|---|
| 1BP/2BP cases | 51 | 43 | 7 | 7 |

**(b)**

| Lin | RH lineage candidates | RH BP | RIVET lineage candidates | RIVET BP |
|---|---|---|---|---|
| XAC | ✓ | ✓ | ✓ | ✓ |
| XAK | Non-recombinant | – | – | ✗ |
| XBB | ✓* | ✗ | – | ✗ |
| XBD | ✓ | ✓ | – | ✓ |
| XBG | ✓ | ✗ | ✓ | ✓ |
| XBJ | ✓ | ✓ | ✓ | ✓ |
| XBL | ✓ | ✓ | ✓ | ✓ |
| XD | ✓ | ✓ | ✓ | ✗ |
| XH | ✓ | ✓ | – | ✗ |
| XM | ✓* | ✓ | ✓ | ✓ |
| XW | ✓ | ✓ | – | ✗ |
| XBR | ✓ | ✓ | ✓ | ✓ |
| XBT | ✓ | ✓ | – | ✓ |
| XBU | ✓ | ✓ | – | ✗ |

**(c)**

| Accession ID | Name | Lineage | #BP in Yeh et al. | #BP in RH | LC in RH |
|---|---|---|---|---|---|
| ON838178.1 | Slovenia/SI2022_S7 | B.1.3 | 1 or 2 | 2: (31–32, 49–50) | B.1.3 + B.1.2 + B.1.3 |
| ON609725.2 | Slovenia/SLO | B.1.3 | 1 or 2 | 2: (31–32, 49–50) | B.1.3 + B.1.2 + B.1.3 |
| ON754985.1 | Slovenia/SI2022_S4 | B.1.3 | 1 or 2 | 2: (31–32, 55–56) | B.1.3 + B.1.2 + B.1.3 |
| ON754986.1 | Slovenia/SI2022_S5 | B.1.3 | 1 or 2 | 2: (31–32, 55–56) | B.1.3 + B.1.2 + B.1.3 |
| ON754987.1 | Slovenia/SI2022_S1_VERDE6 | B.1.3 | 1 or 2 | 2: (31–32, 55–56) | B.1.3 + B.1.2 + B.1.3 |
| ON631241.1 | Slovenia/2022/2 SLO | B.1 | 1 or 2 | 0 | B.1 |
| ON755039.1 | Italy/FVG-ITA_01_2022 | B.1 | 2 | 0 | B.1 |
| ON631963.1 | Australia/VIDRLO1/2022 | B.1 | 2 or 3 | 0 | B.1 |

Given 51 recombinant lineages (1BP and 2BP) according to Pango, RecombinHunt (RH) finds 43 of them in the Nextstrain dataset (see Table 4), while RIVET finds only 7 of them, and finds 7 of them with different donors and acceptors. b Comparison between results of RecombinHunt and RIVET with respect to the Pango lineage ground truth. For both RIVET and RecombinHunt we consider the correctness of the proposed donor and acceptor lineages and the correctness of the breakpoint position (BP). The dash symbol marks the cases where RIVET lineage candidates are not defined in Pango. The asterisk symbol marks the cases in which RH chooses a close ancestor lineage of those indicated in the Pango designation issue. c Results of the recombination analysis on eight mpox sequences. Columns represent the id of sequences and names according to Nextstrain; the lineage assigned by Nextstrain; the information on the number of breakpoints deducible from Yeh et al.[55]; and the results of RecombinHunt in terms of the number of breakpoints (#BP) and acceptor/donor lineage candidates (LC).

We thoroughly evaluated our results in comparison with 51 unambiguous recombinant lineages defined by the Pango SARS-CoV-2 nomenclature for which sufficient genome sequences are available in GISAID. A complete agreement with the ground truth was observed in 40/51 cases (78%). In the majority of incorrectly classified lineages (9 out of 11), recombination was supported only by a relatively low number of target mutations (1 or 2), a scenario that is not incompatible with convergent evolution. When applied to high-quality single sequences, with some explainable exceptions, the method produced highly consistent results with those recovered at the lineage level, and the same correct outcome was reported in the vast majority of the single sequences. Moreover, the small discrepancies observed in our analyses might not necessarily reflect errors and could be suggestive of intra-lineage heterogeneity and/or microevolution in some SARS-CoV-2 recombinant lineages. Note that, for a single lineage (XAT), results obtained on single genome sequences were more in line with the ground truth than the lineage-level consensus-genome.

When applied to real-world data, RecombinHunt outperforms the currently available methods and correctly identifies a significantly larger proportion of recombination events flagged by expert manual analysis in SARS-CoV-2. RecombinHunt correctly flags 33 lineages designated as recombinant by Pango, which however are not identified as recombinant by RIPPLES/RIVET[6,44], hence demonstrating a more than 2fold increase in sensitivity with respect to the current state-of-the-art method for the detection of viral recombination at pandemic/epidemic scale. A direct comparison with GARD[29], 3SEQ[31], or RDP5[34] was not performed, since these methods are not conceived for the analysis of big data and the associated computational requirements do not scale to the analysis of the datasets considered in this work.

Once applied to the monkeypox virus, our method was able to replicate the classification of viral sequences recently indicated as recombinant by using a sophisticated ad-hoc method based on expert manual annotation. A large number of additional candidate recombinant genomes (705 cumulatively) were also detected, suggesting previously unreported recombination events in monkeypox.

Collectively, our results demonstrate that RecombinHunt is highly accurate and reliable, and represents a major breakthrough for the detection of recombinant viruses in large-scale epidemics/pandemics. The method can be applied to most available collections of nucleotide mutations for viral species and facilitates the detection of recombinant viral genomes in current and future viral outbreaks.

## Methods

### Data collection and genome quality filtering

We considered the nucleotide-level mutations of 15,271,031 complete SARS-CoV-2 genome sequences collected from all over the world, downloaded on April 1st, 2023, using EpiCov$^{TM}$ data from the GISAID database[19] available online[59]. The original data included genome sequences in FASTA format and associated metadata (accession ID, collection date, submission date, Pangolin lineage, and collection location). These were processed by the HaploCoV pipeline[27] to derive a large table with the list of mutations and matched metadata for every genome sequence. Only sequences following stringent quality requirements were retained; more specifically, we selected sequences (with ≥ 1 mutation) that hold defined metadata attributes 'Sequence length', 'Type', 'Virus name', and 'Pango lineage'; have 'Is complete' = True; 'Is low coverage' ≠ True; and 'N-Content' ≤2% (according to the definitions given by GISAID). It was also required that sequences did not have conflicting lineage assignments according to HaploCoV and GISAID. The resulting dataset contains 5,255,228 records (c.a. 34.4% of all the available sequences). We also considered the nucleotide-level mutations of 6,983,419 complete SARS-CoV-2 genome sequences collected from the Nextstrain dataset[53]. Here, we downloaded a file containing both metadata and mutations. Then, we retained only the sequences where the metadata attributes 'virus' and 'length' are defined; 'date_submitted' < 31 March 2023; 'QC_missing_data', 'QC_frame_shifts', 'QC_stop_codons', and 'QC_mixed_sites' = 'good'; 'missing_data' ≤2% of the sequence length; 'coverage' ≥99%; and attribute 'QC_overall_status' ≠ 'bad'. The resulting dataset contains 3,984,308 (c.a. 57% of the available sequences).

The implementation was performed with Python v3.10.12. Data filtering and numerical analyses use the numpy (v1.26.0), pandas (v2.1.1), and tqdm (v4.66.1) libraries. Outputs are prepared using, additionally, plotly (v5.17.0), inflect (v6.0.2), tabulate (v0.9.0), and kaleido (0.2.1) libraries. The code is documented on a Zenodo repository[60], available in Jupyter notebooks (prepared using jupyter v1.0.0).

### Mutation-lineage probability

We compute the probability of every genomic mutation in the collection of high-quality SARS-CoV-2 genome sequences defined above 1) in the complete collection; and 2) in each designated Pango Lineage[61]. Probabilities are approximated with the corresponding frequency, i.e., the ratio between the number of genomes holding the mutation and the total number of genomes (1) or the total number of genomes assigned to a lineage (2). Lineages are represented only considering the mutations that are present in a parametric number of sequences of the lineage (75% for SARS-CoV-2). These mutations are called *characteristic mutations*; the list of characteristic mutations for a lineage is denoted as the *lineage mutations-space*. These data are used to estimate the baseline frequency of genomic variants across the complete collection of Pango lineages and in the SARS-CoV-2 genome. In the specific case of SARS-CoV-2 and the phylogenesis-based nomenclature that we use, this results in an 'approximation' of the lineages derived from the phylogenetic tree, removing mutations introduced by recombinant sequences, wrong assignments to lineages, and other noise. Lineages with less than 10 high-quality sequences (i.e., XA, XAQ, XU, and XC in the GISAID dataset) are excluded – to avoid small denominators.

### Screening potential recombinant genomes

The complete workflow of the RecombinHunt method is shown in Fig. 2. The input is a sequence (hereon called *target*) represented as an ordered set of nucleotide mutations, either of an existing genome or of a *consensus-genome*, corresponding to the set of mutations with a frequency above a certain threshold in a given lineage.

The search of candidates is based on the computation of the cumulative logarithmic ratio between the probability of a given mutation $m$ to occur in a given lineage $L_i$ and the probability of that mutation to occur in any SARS-CoV-2 genome. More formally, we associate a function *likelihood_ratio* (see Eq. (1)) to each lineage $L_i$ tested on a target genome $T$ in a range $R$ from a start to an end mutation on $T$:

$$
\text{Likelihood\_ratio}(L_i, T, R_{start:end})
$$
$$
= \sum_{\forall m \in R} \begin{cases} \ln\left(\frac{P(m \in L_i)}{P(m)}\right), & \text{if } m \in T \\ -\ln\left(\frac{P(m \in L_i)}{P(m)}\right), & \text{if } m \notin T \text{ and } m \in L_i \\ 0, & \text{if } m \notin T \text{ and } m \notin L_i \end{cases} \quad (1)
$$

The first candidate – denoted as $L1$ – is searched in the whole space of mutations of the genome, both starting from the 5'-end and the 3'-end. When the high majority of mutations in the target are also included in $L1$ (i.e., all except at most 2) the non-recombinant model is considered as the one that best describes the target. Else, the method evaluates two alternative models, (1) with one breakpoint, separating $L1$ and $L2$, or (2) with double breakpoint, having an $L2$ stretch in between two $L1$ stretches (with a pattern $L1 - L2 - L1$).

In both cases, $L1$ is selected as the first candidate contributing to the largest portion of the genome in the space of mutations of the target; it extends from the 5'-end until the breakpoint ($L1$ has the >> direction) or from the 3'-end to the breakpoint ($L1$ has the << direction). The breakpoint is marked at the mutation where the profile of the $L1$'s *likelihood_ratio* (computed according to) reaches its maximum.

Then, in the case of a single breakpoint, a second candidate, denoted as $L2$ and different from $L1$, is searched in the space of mutations of the genome that was not already covered by $L1$. This case is only considered when there exists an $L2$ (within the Pango lineages) characterized by at least 3 mutations in that space; else, we recede back to the non-recombinant case. The resulting composition of $L1$ and $L2$ also identifies the position of the breakpoint; when the two candidates cover the entire genome space, the breakpoint corresponds to a pair of adjacent positions. Else, we call *gap* the uncovered portion of the genome and propose a gap-resolution procedure. See the below paragraph, for further details.

In the case of a double breakpoint, $L1$ is considered also from the opposite side of the genome (called $L1^{opp}$); if in this region $L1$ has at least 3 mutations of the target, the model $L1 - L2 - L1$ is explored. A second, central candidate $L2$ is then searched between the two stretches of $L1$. After finding $L2$, a gap-resolution procedure may be necessary.

The two models ($L1 - L2$ and $L1 - L2 - L1$) are compared using the Akaike information criterion (AIC). This is framed as follows; we evaluate if $L1 - L2$ (alternative hypothesis) is better than $L1 - L2 - L1$ (null hypothesis). The test obtaining the lowest $p$-value determines the final output.

### Gaps resolution

This procedure is applied when the target positions corresponding to maximum likelihood (i.e., max-$L1$ for $L1$ and max-$L2$ for $L2$) are not adjacent. In such cases, the breakpoint is set as the position $p$ that minimizes the cumulative likelihood loss for both the adjacent regions. The region to the left of the gap is extended up to $p$, while the positions starting from $p + 1$ are assigned to the right-adjacent region.

### Comparison of recombinant vs non-recombinant models

We finally estimate the error probability of the recombination hypothesis against the non-recombination hypothesis. For each target sequence $T$, we compute three *global_likelihood* ($G$) functions, respectively representing the cases where the target is 1) completely

explained by lineage $L_1$ ($G_{L_1}$); 2) completely explained by $L_2$ ($G_{L_2}$); or 3) partially explained by $L_1$ and by $L_2$ in different portions ($G_{L_1,L_2}$) (hence supportive of a recombination). The function cumulatively adds (or subtracts) a contribution for each mutation in the target mutations-space, composed by the combination of $T$ with the involved lineages.

Each term of the sum corresponds to the natural logarithm of the probability of the specific mutation to characterize the lineage $L_i$; the contribution is positive when the mutation occurs on $T$, and negative when it does not. In Eq. (2), we use the general term $L_i$ to represent $L_1$ for $G_{L_1}$ and $L_2$ for $G_{L_2}$. In the composite case of $G_{L_1,L_2}$, instead, $L_i$ is composed of portions of $L_1$ and $L_2$ according to the best 1BP or 2BP solution.

$$
\begin{aligned}
&\text{Global\_likelihood}\,(L_i, T, R_{start:end}) \\
&= \sum_{\forall m \in R, i \in \{T, L_i\}} \begin{cases} \ln(P(m \in L_i)), & \text{if } m \in T \text{ and } m \in L_i \\ -\ln(P(m \in L_i)), & \text{if } m \notin T \text{ and } m \in L_i. \end{cases}
\end{aligned} \quad (2)
$$

Then, by means of AIC, we compare $G_{L_1,L_2}$ versus $G_{L_1}$ and $G_{L_1,L_2}$ versus $G_{L_2}$ and compute $p$-values by based on the relative likelihood. Small $p$-values indicate that the recombinant model $G_{L_1,L_2}$ provides a significantly better fit compared to the single lineage models $G_{L_1}$ and $G_{L_2}$.

### Identification of groups of similar candidate lineage

$L1$ and $L2$ may be represented by additional – similar – candidates. To exhaustively define the group of lineages that are not inconsistent with the roles of, respectively, acceptor and donor, three conditions are checked: 1) the AIC criterion is used to assess how well the stretch of $L1$ (respectively, $L2$) is explained by each of the ten best candidates according to the maximum *likelihood_ratio* value reached in correspondence of the breakpoint position (see Eq. (1)). The candidates from the 2nd to 10th position ones are compared to the first candidate, using the AIC and a hypothesis test with $p$-values $\geq 10^{-5}$, meaning that they are sufficiently different not to be considered acceptable alternatives for (or be differentiated from) $L1$ (resp. $L2$); 2) candidates that reach the maximum of *likelihood_ratio* in locations that are apart from the position of the first candidate (i.e., more than one mutation apart) are not to be considered as acceptable alternatives for $L1$ (resp. $L2$); 3) candidates that do not belong to the same phylogenetic branch or sub-tree as the first candidate are not to be considered acceptable alternatives for $L1$ (resp. $L2$). When a candidate meets all of the three requirements, it is incorporated into a group of alternative candidates, equally explaining the recombination model proposed by RecombinHunt.

### Collection of ground truth information from Pango designation issues

The hunt for recombinants is manually performed by volunteers who report evidence on the Pango designation GitHub repository in the form of issues, broadly documented and discussed with peers[43]. We extracted from the Pango designation file[50] all the entries whose keys start with 'X' at the first level of nomenclature (i.e., without any dot). Then, we matched them to the `lineage_notes.txt` file[62], retrieving the issues where those designations are discussed. We inspected all the issues present on April 1st, 2023.

Two levels of information were recorded: 1) the donor and acceptor lineage candidates of recombination (directly from the `alias_key.json` file[50] and cross-checked with Focosi and Maggi[63], when related information was available); 2) the interval-based position of the breakpoints, manually scouted in the discussions of the issues. As per (1), several candidates[50] are reported with the *name** symbol, indicating that – at the time of designation – it was not possible to assign a precise lineage, but it was possible to assign an entire sub-tree of the phylogeny (with root in *name*). As per (2), sometimes issues' threads reported conflicting intervals; in these cases, we considered

the union of such options. For the purposes of RecombinHunt, intervals in genomic coordinates are translated into target mutations-space coordinates that depend on the target sequence observed in the given task.

### Reporting summary

Further information on research design is available in the Nature Portfolio Reporting Summary linked to this article.

## Data availability

The simulated dataset for sensitivity/specificity analysis generated in this study is provided in Supplementary Data 1. The recombinant genomes identified by RecombinHunt for monkeypox are provided in Supplementary Data 2. Original sequences and metadata of SARS-CoV-2 used in this work are accessible through the GISAID[59] and Nextstrain[54] databases. Original sequences and metadata of mpox used in this work are accessible on the Nextstrain database[57]. The datasets were downloaded on April 1st, 2023. The lists of considered accession IDs are provided on the Zenodo database under accession code https://doi.org/10.5281/zenodo.8123832[60].

## Code availability

RecombinHunt code is provided on Zenodo at https://zenodo.org/doi/10.5281/zenodo.8123832[60]. We include a demo Jupyter notebook and example input/output datasets to reproduce the results presented in the study.

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

## Acknowledgements

We gratefully acknowledge all data contributors, i.e. the Authors and their Originating Laboratories responsible for obtaining the specimens, and their Submitting Laboratories that generated the genetic sequence and metadata and shared via the GISAID Initiative the data on which part of this research is based. We also gratefully acknowledge the Nextstrain team for curating the open SARS-CoV-2 dataset. The work was supported by Ministero dell'Università e della Ricerca (PRIN PNRR 2022 "SENSIBLE" project, n. P2022CNN2J), funded by the European Union, Next Generation EU, within PNRR M4.C2.1.1. Politecnico di Milano, CUP D53D23017400001; Università degli Studi di Milano, CUP G53D23006690001. Principal Investigator A.B., co-Principal Investigator M.C.

## Author contributions

T.A. and A.B. conceived the work; T.A., A.B. and S.C. jointly conceptualized and designed the core algorithm; M.C. performed data preparation, conceptualization of the statistical framework, and data interpretation; T.A. collected the ground truth information, developed the core algorithm, and performed all the experiments; A.B. performed state-of-the-art review, created the figures, and drafted the manuscript; all authors revised the final version of the manuscript; S.C. supervised the project.

## Competing interests

The authors declare no competing interests.
