## [Peer Review File · Nature Communications]

Data-driven recombination detection in viral genomesReviewers' Comments:

Reviewer #1 (Remarks to the Author):

I like reading papers like this, because I like seeing where the new creative computational community is taking various parts of biology. But, I am sorry to say, this paper has ventured into an area of biology (recombination detection) that is very well understood and has more than a dozen methods/approaches published since the mid-1980s. It is rare for a recombination detection method to truly do something new. If you would like a summary of all the recombination-statistic types that have been used since the mid-1980s (first one would be Stephens, 1985, Mol Biol Evol, 2:539) please look in Posada et al (Annu Rev Genetics, 36:75, 2002) and page 78 lists these methods.

1. The key part of this paper is on lines 90-94 where the likelihood approach is described. This site-by-site likelihood approach is similar to the LARD approach (<https://academic.oup.com/mbe/article/16/3/405/2925414>) but one major difference is that the likelihood is based on the likelihood of a site being in a particular state ****based on the collection of sequences you are examining****. This means that a recombinant sequence and two parents can appear in a data set once and show a high likelihood of being recombinant, and then these same three sequences can appear in a different collection and have a different probability of being recombinant. This is not the behavior you want for a recombination detection method.

2. A second feature (common in several recent SARS-CoV-2 recombination detection papers) that goes against the traditional approach of structuring hypotheses is that the phylogenetic tree is built first, which is done under an assumption of clonal non-recombinant evolution. Mutations are then assigned to certain lineages in this tree, and the locations and lineage assignments of locations are used to demonstrate that some sequences are recombinant. But, if some of the sequences are recombinant, then the assumption of clonality is rejected, and you cannot build a simple phylogenetic tree for all the sequences, and this means that the lineage assignments are invalid. I understand that this whole process allows you reject the hypothesis of clonal evolution or non-recombinant evolution. But, you still now need to identify the recombinants themselves, and you can't use the tree's lineage assignments to do it.

3. The authors state:

"Currently, the hunt for recombinant SARS-CoV-2 viral lineages is manually performed by experts who report evidence on the Pango designation GitHub repository in the form of issues; they are broadly documented and allow for discussion with other peers 42 . Here we present a new automatic method (RecombinHunt, Fig. 1) for effectively and efficiently detecting recombination by analyzing the complete data corpus of SARS-CoV-2."

But this is just how Pango does it, and this was done because a manual curation/description/identification of SARS-CoV-2 features was necessary to accompany the large volume of data and analyses that were being produced in 2020 and 2021. We needed some human eyes on these sequence patterns to make sure that the things like novel lineages and recombinants were being identified correctly.

4. The ground truth is simulated sequences, and now of course you can simply download simulated data sets that others have used for these types of analyses. I wouldn't use Pango designations as ground truth, as these are susceptible to the same statistical errors as any other statistical analysis is.

Which brings us to:

5. You can't submit a paper like this without a power analysis. Take a look at Figure 1 in the Posada and Crandall paper from 2001 (PNAS, 98(24):13757-13762), or Figure 2 in the Boni et al 2007 paper (Genetics, 176:1035-1047). These data sets should be available publicly somewhere. If you don't want to use them, you can generate your own of course.

6. Alternatively, or in addition, take a large influenza or ebola data set (negative control), a large SARS-CoV-2 data set (positive control), and see if you can get the expected results with your recombination detection method. You can try also try dengue virus as an in-between example.

7. For figures 1 and 2, extended figure 2: these diagrams are the same diagrams used in forming some of the earlier Posada methods, the simplot method, and probably some others. It would be good to put these in context and explain how your approach differs (if it does).

Reviewer #2 (Remarks to the Author):

Summary

The authors present a likelihood-based approach to identifying individual SARS-CoV-2 sequences as putative recombinants between pairs of a given lineage system, RecombinHunter. It's an interesting and intuitively simple approach, though it suffers from a few fundamental flaws worth noting, such as a dependence on the existence of a high quality, comprehensive lineage system covering extant sequences. Overall, this seems to be a worthy work, particularly if it can be easily adopted by the Pango team and other groups looking to identify recombinant lineage candidates in the latest genetic data.

Major Comments

The authors demonstrate the capability of this approach to categorize individual or consensus sequences from different variants as putative recombinants, but do not explore in practice how one might identify a new, unlabeled or undefined cluster of samples as a putative recombinant lineage. RIPPLES identifies individual branches as recombination event points, and therefore all samples downstream of those events constitute a recombinant lineage; for RecombinHunter to become a useful tool for the Pango team and others to identify novel recombination events, a pipeline for identifying groups of sequences with apparent shared breakpoints putatively reflecting a novel recombinant lineage would be of value. Naturally, this would include a demonstration in practice of this approach identifying a known Pango recombinant lineage from the genetic data available at the original time of designation directly, without any prior labeling of said lineage.

The code is stated to be available on request to the authors. This is inappropriate, as open code

availability is critical for reproduction and review of any algorithm or software toolkit. The authors should upload all code used in this work to a dedicated Github repository and link it in the code availability statement of their manuscript. An example workflow and documentation for use of the code is also critical.

Minor Comments

The authors exclude more than 10 million sequences from their study from concerns regarding sequence quality (Line 85). However, the thresholds and procedure used for filtering is not fully enumerated (define “low coverage”, etc). How is this distinguished from Nextstrain’s curated collection? This should be elaborated on in the Methods section, and the description in the main manuscript can be reduced/replaced with a reference to the Methods section.

The authors perform a comparison with RIPPLES, a phylogeny-based approach to the identification of SARS-CoV-2 recombinants, as applied to the Nextstrain curated set of sequences through the RIVET webui wrapper. They generally find RIPPLES to be insensitive and fail to replicate the Nextstrain recombinants, with only 17 predictions. They do not, however, enumerate the parameters they use for applying RIPPLES/RIVET. It’s possible that an inappropriately high – parsimony-improvement or –num-descendents could explain the general paucity of RIPPLES recombinant predictions. A brief discussion of parameters and tests with alternative parameter sets would be valuable in demonstrating the relative advantage of RecombinHunter over RIPPLES.

On line 169, the authors make reference to insensitivity “due to attraction to local maxima and limited/insufficient drops in the cumulative likelihood ratio score” but do not expand on this potential weakness. Elaboration on this point, including examples, would be valuable for understanding the limitations of this method.

Line Comments

Line 24- add citation for XE growth advantage

Line 25- “the recombinant XBB lineage, the ancestor of XBB.1.5 and XBB.1.16 is the only Variants Of Interest” -> “the recombinant XBB lineage is the only Variant of Interest”. The naming of XBB.1.5 and XBB.1.16 implies the XBB ancestry, so it doesn’t need to be restated.

Line 83- why 75% for the critical threshold? How would changing this value affect the results?

Line 112- why 3 mutations for the 2BP case? Two mutations largely unique to L1 in the L1opp area surely could also shift the model towards a 2BP explanation. Perhaps this and similar thresholds could just be a parameter set by a user in the software.

Line 121- missing citation for the AIC

Reviewer #3 (Remarks to the Author):

Alfonsi T, et al. developed the RecombinHunt, an algorithm for detecting virus recombinant lineages using large amount of viral genome sequences deposited on a sequence repository such as GISAID. The authors tested the algorithm with a set of assigned Pango distinct SARS-CoV-2 recombinant lineages to demonstrate the algorithm's accuracy in detecting recombination breakpoints.

Moreover, they showed that the algorithm can outperform some of the existing tools like RIVET. Finally, the authors showed the algorithm's ability to detect some novel recombinant lineages when applied to the monkeypox virus. However, some analytical procedures and terminology issues still need to be addressed/clarified.

The authors stated that "Our approach relies exclusively on big data-driven methods"? Also, the author mentioned that "The method can be applied to any mid-size available collections of nucleotide mutations for viral species". It will be helpful to provide a precise definition for big-data and mid-size in the context of a pandemic i.e. big data is not available at the early stage of a pandemic?

It is important to clarify the type of recombination that is detectable by the algorithm? Precise (homologous) or imprecise (nonhomologous). Can RecombinHunt detect deletions and insertions, which have been shown to play a critical role in shaping the pandemic?

Does RecombinHunt differentiate between different kinds of mutations, which can influence the accuracy of calculating the breakpoints? For example, it was found that SARS-Cov-2 is biased towards the mutation C to U.

Did the authors look at how the location of the recombination breakpoint would impact the algorithm's performance? Some locations can be under stronger selection pressure than others, and others can be more recombinogenic. These facts can influence the accuracy of calling recombination junction points.

Related to the previous comment, the authors tested the algorithm using a set of high-quality and well-defined recombinants. It will be useful if the author can also test the algorithm against a set of simulated datasets that contain different recombination breakpoints distributed all over the genome from different viruses. This can provide a more robust and inclusive way of testing the accuracy of the algorithm.

The author stated in the abstract that "several methods for detecting recombination in SARS-CoV-2 have been proposed; however, none could faithfully reproduce manual analyses by experts in the field". How was the reproducibility tested for the other methods?

The author stated that "the method produces results that are easily inspectable on simple visual reports...thus it is completely transparent to its users". What does "transparent" mean in this context and how can this be translated upon comparing with other methods?

.

What is the x-axis in Fig 1 and 2? need clarification

Point-by-point response to the reviewers' comments_w

We thoroughly revised our manuscript according to the indications of the reviewers.

In particular, we added a new section discussing sensitivity (by adopting the methodology used by RIVET/RIPPLES), specificity (false positives rate), and minimum requirements of applicability of our method. Based on these results, we confirm that RecombinHunt can be applied to any virus for which a structured phylogeny/classification system is available, this includes several viral pathogens for which curated analyses are maintained by the Nextstrain community, including SARS-CoV-2, Monkeypox, Dengue, RSV, Influenza, Enterovirus D68, and the West Nile virus.

We also made more explicit our comparison with the competing method RIVET/RIPPLES, by adding a summary table showing that our method captures 43 out of 51 manually curated ground truth cases of Pangolin while RIVET/RIPPLES misses 33 cases out of 51 and is not superior to us in the 14 overlapping cases.

We believe that the addition of the new section on sensitivity, specificity, and minimum requirements, as requested by all reviewers, has greatly strengthened our paper, by indicating its broad applicability. Moreover, reviewers pointed out aspects of our paper that required more extensive descriptions - these contributions are highlighted in red color in our revised manuscript, whose main results and methods have not changed. We are confident that, after revision, our manuscript meets the high-quality standards of Nature Communications.

Reviewer #1

(Remarks to the Author):

I like reading papers like this, because I like seeing where the new creative computational community is taking various parts of biology. But, I am sorry to say, this paper has ventured into an area of biology (recombination detection) that is very well understood and has more than a dozen methods/approaches published since the mid-1980s. It is rare for a recombination detection method to truly do something new. If you would like a summary of all the recombination-statistic types that have been used since the mid-1980s (first one would be Stephens, 1985, Mol Biol Evol, 2:539) please look in Posada et al (Annu Rev Genetics, 36:75, 2002) and page 78 lists these methods.

We highly respect the background of the reviewer, who is citing important works of the mid-1980s and later. In SARS-CoV-2-related research, we have widely used our big data approach, stemming from an ERC AdG on genomic computing (S. Ceri, 2016-2021). Along with the methods of big data, we have published recent works on SARS-CoV-2 in journals such as Nucleic Acid Research ([doi:10.1093/nar/gkaa846](https://doi.org/10.1093/nar/gkaa846), [doi:10.1093/nar/gkab478](https://doi.org/10.1093/nar/gkab478)), Molecular Biology and Evolution ([doi:10.1093/molbev/msab049](https://doi.org/10.1093/molbev/msab049)), Communications Biology ([doi:10.1038/s42003-023-04784-4](https://doi.org/10.1038/s42003-023-04784-4)), GigaScience ([doi:10.1093/gigascience/qiad036](https://doi.org/10.1093/gigascience/qiad036)), Scientific Data ([doi:10.1038/s41597-022-01348-9](https://doi.org/10.1038/s41597-022-01348-9)), Scientific Reports ([doi:10.1038/s41598-021-00496-z](https://doi.org/10.1038/s41598-021-00496-z)), Database ([doi:10.1093/database/baab059](https://doi.org/10.1093/database/baab059), [doi:10.1093/database/baad044](https://doi.org/10.1093/database/baad044)); many of them were summarized in a recent keynote on “Big data in life sciences, from theory to applications”, given by S. Ceri at Informatics Europe in Edinburgh (Nov. 2023) [Slides].

Of course, we are aware of Stephens et al. and Posada et al. works; however, we compared our work with more recent approaches that are the most competitive ones in the context of large data collections and adopted, like us, a big data approach. In particular, the papers on RIPPLES (published in Nature in 2022) and RIVET (available as a preprint on bioRxiv at the time of our submission, and recently published on Bioinformatics) also presented a big data method, based on phylogeny; we thoroughly compare the results produced by our method with the results obtained using the RIVET software. We used as input the same dataset (downloading sequences from the Nextstrain database, retrieved on the date of their analysis). In summary, our method perfectly captures 43 out of 51

manually curated ground truth cases of Pangolin, whereas RIVET/RIPPLES captures only 14 of them, and – in 7 of these cases – it designates donor/acceptor lineages not defined in Pangolin.

Thus, we believe that our method, based on mutation frequencies, is a strong contribution to the detection of viral recombinations—it uses in an accurate way a body of information that is available to the scientific community at large through open datasets. Thanks to our analysis on 'Sensitivity, specificity, minimum requirements' (Pages 6–7, lines 126–172 in the revised manuscript), as rightfully requested by all reviewers, we have now stronger arguments that show how our method applies to many viral species in addition to SARS-CoV-2 and Monkeypox, provided that a dataset with a well-defined set of lineages characterized by defining mutations is available – as we trust it will be the case for future epidemics and pandemic threats. Besides recombination, we also used big data for the early detection of “variants” (lineages with strong prevalences, see [doi:10.1093/database/baad044](https://doi.org/10.1093/database/baad044) and [doi:10.1038/s41598-021-00496-z](https://doi.org/10.1038/s41598-021-00496-z)) and we are currently working on methods for detecting reassortment in viral species (most noticeably, Influenza viruses).

We sincerely hope the reviewer will judge our work based on the above premises. Below, we provide point-to-point responses to the comments in the review.

1. The key part of this paper is on lines 90-94 where the likelihood approach is described. This site-by-site likelihood approach is similar to the LARD approach (<https://academic.oup.com/mbe/article/16/3/405/2925414>) but one major difference is that the likelihood is based on the likelihood of a site being in a particular state ****based on the collection of sequences you are examining****. This means that a recombinant sequence and two parents can appear in a data set once and show a high likelihood of being recombinant, and then these same three sequences can appear in a different collection and have a different probability of being recombinant. This is not the behavior you want for a recombination detection method.

From this comment we realized that there remain two points of misunderstanding with the reviewer:

- a) Methods developed for single sequence analysis are not applicable when big genome datasets are considered (else, they would have been employed with SARS-CoV-2 itself). The LARD approach – as well as Boni et al. 2007 ([doi:10.1534/genetics.106.068874](https://doi.org/10.1534/genetics.106.068874)) and Lam et al. 2018 ([doi:10.1093/molbev/msx263](https://doi.org/10.1093/molbev/msx263)) – analyzes triplets of sequences; instead, in RecombinHunt likelihood is used to compute the similarity of ONE candidate sequence with MANY existing lineages along the viral genome. Then, a direct comparison between our method and LARD seems not to be appropriate.*
- b) We do not recognize our approach as descending from the LARD approach, in which “candidate parental sequences should be readily identifiable from the prior split decomposition analysis” (end of Column 3 of the PDF); we do not perform any split analysis.*

2. A second feature (common in several recent SARS-CoV-2 recombination detection papers) that goes against the traditional approach of structuring hypotheses is that the phylogenetic tree is built first, which is done under an assumption of clonal non-recombinant evolution. Mutations are then assigned to certain lineages in this tree, and the locations and lineage assignments of locations are used to demonstrate that some sequences are recombinant. But, if some of the sequences are recombinant, then the assumption of clonality is rejected, and you cannot build a simple phylogenetic tree for all the sequences, and this means that the lineage assignments are invalid. I understand that this whole process allows you reject the hypothesis of clonal evolution or non-recombinant evolution. But, you still now need to identify the recombinants themselves, and you can't use the tree's lineage assignments to do it.

As noted by the reviewer, we rely on the Pango nomenclature for the classification of SARS-CoV-2 lineages. However, we do not directly use mutation assignments to lineages as they are. Our characteristic mutations are re-computed considering the mutations that are present in a parametric number of sequences of the lineage (75% for SARS-CoV-2); this results in an ‘approximation’ of the

lineages derived from the tree, removing mutations introduced by recombinant sequences, wrong assignments to lineages, and other noise.

Note that our method would work also with any alternative classification of sequences (possibly, agnostic with respect to lineages); we chose to adapt to Pango lineages only because this is the de facto standard in the SARS-CoV-2 community, and it allowed us to make comparisons with manually curated results and with competitor methods.

3. The authors state:

"Currently, the hunt for recombinant SARS-CoV-2 viral lineages is manually performed by experts who report evidence on the Pango designation GitHub repository in the form of issues; they are broadly documented and allow for discussion with other peers ⁴². Here we present a new automatic method (RecombinHunt, Fig. 1) for effectively and efficiently detecting recombination by analyzing the complete data corpus of SARS-CoV-2."

But this is just how Pango does it, and this was done because a manual curation/description/identification of SARS-CoV-2 features was necessary to accompany the large volume of data and analyses that were being produced in 2020 and 2021. We needed some human eyes on these sequence patterns to make sure that the things like novel lineages and recombinants were being identified correctly.

We understand the reviewer's defense of manual curation and of the Pango system, in the specific context of the SARS-CoV-2 pandemic. Let us point out that our method confirms most findings of the Pango community, and it does it on the ground of solid computations. As RecombinHunt was able to identify most recombinations spotted by "human eyes", we can conclude that our method is accurate. Notably, we identified controversial cases where the "human eye" called recombination events that are of controversial nature (see the discussion on assignments in the paper – Page 8, lines 192–216 – and in Supplementary File 5).

4. The ground truth is simulated sequences, and now of course you can simply download simulated data sets that others have used for these types of analyses. I wouldn't use Pango designations as ground truth, as these are susceptible to the same statistical errors as any other statistical analysis is.

In agreement with the reviewer's suggestions, we applied our method on simulated data (available in Supplementary Dataset 1, see the new Section 'Sensitivity, specificity, minimum requirements') and our outcomes showed higher sensitivity levels w.r.t. the results obtained on the real datasets – this is consistent with expectations.

When analyzing the real datasets, instead, we chose to use Pango designations as 'ground truth' as it represents the de facto standard and it can be used to compare our method to others. In these regards, we would like to stress that RecombinHunt did correctly identify a significantly larger number of recombinant lineages, determined by manual analyses by experts, compared with the state of the art in the field.

Which brings us to:

5. You can't submit a paper like this without a power analysis. Take a look at Figure 1 in the Posada and Crandall paper from 2001 (PNAS, 98(24):13757-13762), or Figure 2 in the Boni et al 2007 paper (Genetics, 176:1035-1047). These data sets should be available publicly somewhere. If you don't want to use them, you can generate your own of course.

All reviewers have requested us to perform sensitivity, specificity, and minimum requirement analysis on simulated data. We responded to this request by adding a new section to the paper (Pages 6–7,

lines 126–172). For sensitivity and specificity, we generated our datasets using the approach described in RIVET/RIPPLES, although adapted to our space, which considers mutations and their cumulative frequencies rather than phylogeny.

For defining the minimum requirements of our method, we used 1) an analytical approach based on the binomial distribution and hypothesis testing within the mutations-space, and 2) random re-sampling from real datasets of SARS-CoV-2 and Monkeypox, as each viral species has specific peculiarities in their mutation distribution.

6. Alternatively, or in addition, take a large influenza or ebola data set (negative control), a large SARS-CoV-2 data set (positive control), and see if you can get the expected results with your recombination detection method. You can try also try dengue virus as an in-between example.

Based on this comment from the reviewer, we now include a ‘Specificity’ analysis (Page 6–7, lines 138–145), to estimate the False Positives Rate of our method, using a non-recombinant SARS-CoV-2 lineage. We preferred not to use Influenza datasets as -in that case- the absence of recombination is debated (e.g., [doi:10.1016/j.gene.2011.04.012](https://doi.org/10.1016/j.gene.2011.04.012), [doi:10.1016/j.gene.2011.10.041](https://doi.org/10.1016/j.gene.2011.10.041)).

7. For figures 1 and 2, extended figure 2: these diagrams are the same diagrams used in forming some of the earlier Posada methods, the simplot method, and probably some others. It would be good to put these in context and explain how your approach differs (if it does).

Indeed, our method was inspired by the plots in Fig. 1 of Boni et al. 2007 and Fig. 1 of Lam et al. 2018. These methods only consider three (given) sequences; the two figures in those works represent a sequence of positive/negative steps corresponding to the equality of the sequence (informative) positions with either of the two parents. All steps are equally weighted, and their arrangement defines a random variable (i.e., a Hyper Geometric Random Walk) whose probability corresponds to the probability of a recombination event. Differently, RecombinHunt (i) uses quantitative information to assign a weight to every position and candidate lineage, (ii) finds the most likely lineage assignments, and (iii) computes the likelihood of a recombination event according to our own formula (see Methods). We would like to clarify that the plots in Figures 1 and 2 (which correspond to Figures 1 and 3 in the revised version of the manuscript) are not random walks but actual likelihood values for the considered candidates.

Reviewer #2

(Remarks to the Author):

Summary

The authors present a likelihood-based approach to identifying individual SARS-CoV-2 sequences as putative recombinants between pairs of a given lineage system, RecombinHunter. It's an interesting and intuitively simple approach, though it suffers from a few fundamental flaws worth noting, such as a dependence on the existence of a high quality, comprehensive lineage system covering extant sequences. Overall, this seems to be a worthy work, particularly if it can be easily adopted by the Pango team and other groups looking to identify recombinant lineage candidates in the latest genetic data.

We would like to thank the reviewer for this positive assessment of our work.

Major Comments

The authors demonstrate the capability of this approach to categorize individual or consensus sequences from different variants as putative recombinants, but do not explore in practice how one might identify a new, unlabeled or undefined cluster of samples as a putative recombinant lineage. RIPPLES identifies individual branches as recombination event points, and therefore all samples downstream of those events constitute a recombinant lineage; for RecombinHunter to become a useful tool for the Pango team and others to identify novel recombination events, a pipeline for identifying groups of sequences with apparent shared breakpoints putatively reflecting a novel recombinant lineage would be of value. Naturally, this would include a demonstration in practice of this approach identifying a known Pango recombinant lineage from the genetic data available at the original time of designation directly, without any prior labeling of said lineage.

The purpose of our method is to determine recombinants of given lineages; clustering sequences into lineages is a conceptually different task. Our method can also be applied to individual sequences (see Section "Analysis of single high-quality sequences"), so any decision-making body could set up rules about when a recombinant lineage should be called based on the number of similar sequences from the same donor/acceptor pair (e.g., as it currently happens in Pango designations). Methods for aggregating genomic sequences based on sequence similarity are already available. One of the authors proposed the HaploCoV method as an alternative classification based on big data (M. Chiara, Communications Biology, [doi:10.1038/s42003-023-04784-4](https://doi.org/10.1038/s42003-023-04784-4)) and the two methods (HaploCov and RecombinHunt) could be integrated, but then -in any case- results would be subject to a manual/expert curation and a manual placement into the phylogenetic tree.

The code is stated to be available on request to the authors. This is inappropriate, as open code availability is critical for reproduction and review of any algorithm or software toolkit. The authors should upload all code used in this work to a dedicated Github repository and link it in the code availability statement of their manuscript. An example workflow and documentation for use of the code is also critical.

Our code is public and documented ([doi:10.5281/zenodo.10154093](https://doi.org/10.5281/zenodo.10154093)). We include a demo Jupyter notebook and example input/output datasets to reproduce the results presented in the study. This had been specified in the letter to the editor; we deeply apologize to all reviewers for not providing it directly in the manuscript at the time of the original submission.

Minor Comments

The authors exclude more than 10 million sequences from their study from concerns regarding sequence quality (Line 85). However, the thresholds and procedure used for filtering is not fully enumerated (define “low coverage”, etc). How is this distinguished from Nextstrain’s curated collection? This should be elaborated on in the Methods section, and the description in the main manuscript can be reduced/replaced with a reference to the Methods section.

In agreement with the comment of the reviewer, we removed this information from the main text, now referencing the Methods section in two points (see Page 4, line 69 and see Page 11, line 257). In the Methods, the paragraph on the GISAID data processing pipeline has been extended (see Page 27, lines 468–472), by explicitly stating which metadata were derived from GISAID and which values were used for filtering. A paragraph has been added (see Page 27, lines 473–479) to describe also the processing pipeline for the Nextstrain dataset.

The authors perform a comparison with RIPPLES, a phylogeny-based approach to the identification of SARS-CoV-2 recombinants, as applied to the Nextstrain curated set of sequences through the RIVET webui wrapper. They generally find RIPPLES to be insensitive and fail to replicate the Nextstrain recombinants, with only 17 predictions. They do not, however, enumerate the parameters they use for applying RIPPLES/RIVET. It’s possible that an inappropriately high –parsimony-improvement or –num-descendents could explain the general paucity of RIPPLES recombinant predictions. A brief discussion of parameters and tests with alternative parameter sets would be valuable in demonstrating the relative advantage of RecombinHunter over RIPPLES.

We would like to clarify that we did not run RIVET: we just downloaded from their tool the most updated result at the time of writing and applied our method to the same dataset, i.e., Nextstrain data as of March 30, 2023.

On line 169, the authors make reference to insensitivity “due to attraction to local maxima and limited/insufficient drops in the cumulative likelihood ratio score” but do not expand on this potential weakness. Elaboration on this point, including examples, would be valuable for understanding the limitations of this method.

We thank the reviewer for this comment, which has given us an indication of how some of our explanations about insensitivity were hard to understand – they descend from our habit of reading charts of cumulative likelihood, as observed, e.g., in the visual analyses associated with Pango recombinations computed using GISAID and Nextstrain data (see Supplementary Files 1–4). In the revised manuscript, we accordingly expanded our discussion and explained in more readable terms our conjecture (see Page 9, lines 211–216).

Line Comments

Line 24- add citation for XE growth advantage

This has been added (see Page 2, line 23).

Line 25- “the recombinant XBB lineage, the ancestor of XBB.1.5 and XBB.1.16 is the only Variants Of Interest” -> “the recombinant XBB lineage is the only Variant of Interest”. The naming of XBB.1.5 and XBB.1.16 implies the XBB ancestry, so it doesn’t need to be restated.

This has been corrected (see Page 2, lines 23–25).

Line 83- why 75% for the critical threshold? How would changing this value affect the results?

The chosen threshold is also set by other well-established resources in the community (e.g., [Outbreak.info](https://outbreak.info), [doi:10.1038/s41592-023-01769-3](https://doi.org/10.1038/s41592-023-01769-3)). Our new section on 'Minimum requirements' connects, from a statistical perspective, thresholds to other variables defined for each lineage (see Page 7, lines 146–161).

Line 112- why 3 mutations for the 2BP case? Two mutations largely unique to L1 in the L1opp area surely could also shift the model towards a 2BP explanation. Perhaps this and similar thresholds could just be a parameter set by a user in the software.

In our opinion, when just two mutations occur in the viral population with relatively high frequencies, convergent evolution, and positive selection could represent an equally plausible alternative model to recombination. However, we agree with the reviewer that this choice should be given a parameter in RecombinHunt; the software made available on Zenodo ([doi:10.5281/zenodo.10154093](https://doi.org/10.5281/zenodo.10154093)) allows users to change this threshold.

Line 121- missing citation for the AIC

This has been added (Page 5, line 114).

Reviewer #3

(Remarks to the Author):

Alfonsi T, et al. developed the RecombinHunt, an algorithm for detecting virus recombinant lineages using large amount of viral genome sequences deposited on a sequence repository such as GISAID. The authors tested the algorithm with a set of assigned Pango distinct SARS-CoV-2 recombinant lineages to demonstrate the algorithm's accuracy in detecting recombination breakpoints. Moreover, they showed that the algorithm can outperform some of the existing tools like RIVET. Finally, the authors showed the algorithm's ability to detect some novel recombinant lineages when applied to the monkeypox virus.

We agree on this assessment and thank the reviewer for the following comments, to which we respond below.

However, some analytical procedures and terminology issues still need to be addressed/clarified.

The authors stated that "Our approach relies exclusively on big data-driven methods"? Also, the author mentioned that "The method can be applied to any mid-size available collections of nucleotide mutations for viral species". It will be helpful to provide a precise definition for big-data and mid-size in the context of a pandemic i.e. big data is not available at the early stage of a pandemic?

We agree with the reviewer that these descriptions were not appropriately precise. We then eliminated all mentions of "big" and "mid-size" data and added a rigorous 'Minimal requirements' analysis (Page 7, lines 146–172). For what concerns the "minimal data size" we show that the method is successful in confirming/discovering Monkeypox recombinants using 5,402 sequences, most of them collected between May 2022 and May 23, 2023.

It is important to clarify the type of recombination that is detectable by the algorithm? Precise (homologous) or imprecise (nonhomologous). Can RecombinHunt detect deletions and insertions, which have been shown to play a critical role in shaping the pandemic?

RecombinHunt uses mutations at the nucleotide level, including deletions and insertions, as determined by standard variant calling methods. These mutations are not determined by RecombinHunt itself but are taken as input to our method; they are determined based on standard variant calling workflows. Based on these premises, we believe that the method can only identify precise homologous recombination since all the computations are performed on the top of a reference genome.

Does RecombinHunt differentiate between different kinds of mutations, which can influence the accuracy of calculating the breakpoints? For example, it was found that SARS-Cov-2 is biased towards the mutation C to U.

As mentioned in the answer to the previous point, RecombinHunt does not infer mutations; instead, it considers as its input the set of mutations of a sequence, as determined by variant calling methods (specifically, using HaploCoV, which employs MUMmer ([doi:10.1371/journal.pcbi.1005944](https://doi.org/10.1371/journal.pcbi.1005944))). We describe a mutation in terms of its position and type (substitution, insertion, and deletion) and then we build our statistics by using this information as input.

Did the authors look at how the location of the recombination breakpoint would impact the algorithm's performance? Some locations can be under stronger selection pressure than others, and

others can be more recombinogenic. These facts can influence the accuracy of calling recombination junction points.

We would like to clarify that our method does not directly align and compare genome sequences, but instead, genomes are represented as a set of mutations, as determined by a variant calling algorithm. It follows that all our computations are not performed on genomic coordinates; rather, by considering the ordered set of mutations in a target sequence.

Nevertheless, we noticed that – in the case of lineages XAW, XAT, and XP (group G3, Page 9, from line 205) – when recombination events are supported only by a limited number of mutations, 1) localized at the terminal ends of the genome and 2) associated with a relatively high frequency in the viral population, our method might suffer a loss of sensitivity. A sentence has been added to the revised manuscript to clarify this point (see Page 9, lines 211–216).

Related to the previous comment, the authors tested the algorithm using a set of high-quality and well-defined recombinants. It will be useful if the author can also test the algorithm against a set of simulated datasets that contain different recombination breakpoints distributed all over the genome from different viruses. This can provide a more robust and inclusive way of testing the accuracy of the algorithm.

This request was shared by all reviewers; in response to it, we generated simulated datasets by adding non-characteristic mutations as noise, and we showed an extremely high sensitivity and specificity of the method. See the new “Sensitivity, specificity, minimum requirements” section of the revised manuscript (Pages 6–7, lines 126–172).

The author stated in the abstract that "several methods for detecting recombination in SARS-CoV-2 have been proposed; however, none could faithfully reproduce manual analyses by experts in the field". How was the reproducibility tested for the other methods?

We thank the reviewer for spotting our misuse of the term “reproducible”; here, we just meant to say that no other method could obtain accurate outcomes as our method. We amended the sentence in the abstract by replacing ‘reproduce’ with ‘confirm’. Coherently, we also changed the name of the subsection “Reproducibility on individual high-quality sequences” into “Analysis of single high-quality sequences” (Page 9, line 217) as we agree this is less misleading.

The author stated that "the method produces results that are easily inspectable on simple visual reports...thus it is completely transparent to its users". What does "transparent" mean in this context and how can this be translated upon comparing with other methods?

Please consider the visual analyses that we associated with all Pango recombination cases, determined by using either GISAID data (Supplementary File 2) or Nextstrain data (Supplementary File 3). Having all this information condensed into a single report explains all the aspects of the application of the method. We used the term “transparent” as opposed to “black-box computations”, which hide the parameters and results of their analyses.

However, we agree with the reviewer that this statement could be misinterpreted, so we removed the sentence (previously on Page 4).

What is the x-axis in Fig 1 and 2? need clarification

Consider Figure 1 first, as a sort of “graphical abstract” of the method, which is explained in the ‘Approach’ section. Here, our x-axis is the “mutations-space” for three target genomes.

The first space goes from 1 to $N_1 = 66$, i.e., the mutations present in the first target non-recombinant genome. RecombinHunt fails to find any breakpoint and thus associates the target to the BA.2.3.13

lineage.

The second space goes from 1 to $N_2 = 72$, i.e., the mutations present in the second target genome with one breakpoint. RecombinHunt assigns it to the XBE Pango Lineage, a 1BP recombinant case with BA.5.2.6 as the first candidate and BE.4 as the second candidate.

The third space goes from 1 to $N = 67$, i.e., the mutations present in the third target genome with 2 breakpoints. RecombinHunt assigns it to the XD Pango Lineage, a 2BP recombinant case with AY.4 as the first candidate (both from left and right) and BA.1.22 as the second candidate (center).

In panel (b), we identify the breakpoint in the mutations-space of the second sequence between mutations 63 and 64, where the cumulative maximum likelihood score between the target sequence and the best candidate lineage L1 associated with the target has a considerable drop; we then determine a second lineage L2 as the best candidate lineage associated with the target from the opposite end.

After this premise, it becomes clear that mutation points (of the mutations-space) and the actual positions in the viral genome are in the same sequence, but the genomic distance between any two points (i.e., mutations) is not fixed.

In response to a previous point of the reviewer, we would like to note that “selection pressure” or “recombinogenicity” of genomic locations does influence the density of mutations in the genome; however, in our space, which lists each mutation individually, these aspects do not influence the accuracy of determining breakpoints, as each position gives its own specific and independent contribution. Indeed, in panel (b) the breakpoint is expressed as “the interval between the 63rd and 64th positions of the mutations of the target”; these can be translated to coordinates in the viral genome (we provide this additional information in the complete analyses attached as Supplementary Files 2 and 3).

Figure 2 (as asked by the reviewer), which has become Figure 3 in the revised manuscript, is very similar, but it shows consensus genomes made of characteristic mutations, with a threshold set to 75%. Thus, panel (a) represents the lineage XBE, with 72 mutations, whereas panel (b) represents the lineage XD, with 67 mutations.

REVIEWER COMMENTS

Reviewer #1 (Remarks to the Author):

OK, I see that the authors have made a lot of additions and they have provided a lot of explanations in the reviewer comments. I am still not convinced of a few things, and I will describe this here. Based on the other two reviews, this may end up being an accept, so please make sure the issues below are addressed.

#1. This is a mosaic method. Please state so simply in the paper. You are computing a likelihood ratio of a 2-breakpoint recombinant to a direct non-recombinant descendant, or the likelihood ratio of a 1-breakpoint recombinant to a direct non-recombinant descendant. This is in fact similar to the LARD approach (one sequence's evolutionary origin is evaluated by the likelihood ratio tests above) but in the LARD approach the query sequence is only compared against two other sequences whereas in RecombinHunt the query sequence is compared against many different lineages.

I understand that your paper does not do split decomposition. The LARD method does not do this either. The 1999 MBE paper (vol 16, no 3, pp.405-409, titles "Phylogenetic Evidence for Recombination in Dengue Virus") simply includes a split decomposition approach to help identify which parental sequences to test against. Look at the left column of page 406 of that paper. The long paragraph in the middle describes their likelihood approach which seems to be the same as the one in your paper. There's nothing wrong with that. But you should say where the approach comes from.

#2. The "Rejected Hypothesis Tree Problem": if you reject the hypothesis of no recombination, you can't build a tree. I appreciate that you now see this as a problem. And I liked parts of your response in the author rebuttal. You said:

"Our characteristic mutations are re-computed considering the mutations that are present in a parametric number of sequences of the lineage (75% for SARS-CoV-2); this results in an 'approximation' of the lineages derived from the tree, removing mutations introduced by recombinant sequences, wrong assignments to lineages, and other noise."

This might be a good approach to building an appropriate backbone tree, even when you have recombination, so that you can test recombinants against this backbone tree and all its lineages. But this method is not described anywhere in the ms. I appreciate that this is a very tall problem to solve, and that none of the lineage-assignment-recombination methods have solved this problem yet.

If the other two referees are willing to drop point #2, I will also drop my criticism here, but this open problem in "recombination detection via lineage assignment" needs a paragraph in the discussion.

#3. In the new power analysis, you generated recombinant sequences and added noise by adding 0, 1, 2, or 3 mutations. This is not very many mutations, and any recombination method should be able to detect these recombinants. This is why you are getting 99% statistical power in this simulated data set. In a power analysis, you are supposed to compare RecombinHunt to GARD, 3SEQ, RDP, RIPPLES and show that RecombinHunt has similar or better power, similar or better specificity than

these other methods. The specificity analysis on page 6 looks good to me.

I am very sorry to say this, and I really like the effort you have put into the revision, but a power analysis without other methods present is not good enough for a journal like Nature Communications. Please ask referees 2 and 3 if they agree with me.

Reviewer #2 (Remarks to the Author):

Summary

I appreciate the author's efforts in addressing reviewer comments in developing a specificity and sensitivity analysis. This manuscript is notably improved.

Major Comments

I believe the authors may have misunderstood one of my prior major comments- I apologize for the miscommunication. I do not expect RecombinHunter to identify novel lineages from a phylogeny or generic sequence dataset; rather, the idea is to decide whether a small group of sequences identified via some other method is a recombinant of two established lineages. The authors do recognize how identification of donor/acceptor pairs can be part of this workflow in their response, and I would like to see this discussed and/or briefly exemplified in the work proper. I would particularly appreciate an explicit demonstration that RecombinHunter could have identified XBB and other lineages descended from a novel recombinant event accurately given only the data available at the time. The authors do of course analyze the efficacy of RecombinHunter as applied to all identified recombinant sequences and lineages in the wider dataset, but this is a slightly different problem from examining the theoretical historical performance of this approach in practice. This extension of the author's analysis, while not ultimately essential, would make it very clear how RecombinHunter is a valuable addition to the toolkit of genomic epidemiologists and lineage curators.

Minor Comments

I appreciate the authors clarifying that RIVET results were downloaded directly from the portal. I am mildly concerned about the apparent insensitivity of RIVET's RIPPLES parameters, though it does avoid needing a curated lineage system as a starting point. Naturally, changes to RIVET are far beyond the scope of this work or the responsibility of the authors, though I would appreciate a brief statement that the insensitivity may be due to the specific configuration underlying the web portal rather than inherent to the statistical approach.

Reviewer #3 (Remarks to the Author):

The authors did a commendable job of addressing my major comments. They took my suggestions into account and implemented them effectively. The revisions made were well-thought-out and contributed significantly to the overall quality of their work. It is evident that the authors are dedicated to producing a high-quality piece, and I applaud their efforts.

Point-by-point response to the reviewers' comments

We revised our manuscript according to the indications of the editor and the reviewers. We believe that all comments contributed to strengthening our manuscript and clarifying why RecombinHunt can be a valuable addition to the toolkit of genomic epidemiologists and lineage curators (as suggested by reviewer 2).

New contributions are highlighted in red color in our revised manuscript, whose main results and methods have not changed substantially. With respect to the previous version you will find that we:

- Improved our sensitivity analysis by simulating higher mutation rates (0, 3, 5, 10, 15, 20, 30 simulated mutations) on batches of 500 sequences. Please, note that inserting 30 mutations corresponds to adding noise of 50% with respect to a "typical" recombinant sequence. Accordingly, we also refined our previous specificity analysis by switching to batches of 500 sequences (instead of 250 sequences used previously).*
- Included a new section on the predictive power of RecombinHunt entitled: "Post hoc detection of recombinant lineages", where we assess RecombinHunt's ability to correctly label recombinant sequences that were later assigned to recombinant lineages in the Pango nomenclature.*
- Added a paragraph in the discussion to highlight more explicitly the limitations of our approach.*
- Added a short paragraph in the discussion to explain why we are not in the position to replicate the sensitivity analysis performed on RecombinHunt by using other methods.*

Reviewer #1

(Remarks to the Author):

OK, I see that the authors have made a lot of additions and they have provided a lot of explanations in the reviewer comments. I am still not convinced of a few things, and I will describe this here. Based on the other two reviews, this may end up being an accept, so please make sure the issues below are addressed.

We appreciate the effort of the reviewer in reassessing our work in view of the additions to the manuscript and explanations provided in the rebuttal. The reviewer's comments – in the previous round of review and especially in this second round – have allowed us to improve the presentation of our contribution. We respond specifically to each point below.

#1. This is a mosaic method. Please state so simply in the paper. You are computing a likelihood ratio of a 2-breakpoint recombinant to a direct non-recombinant descendant, or the likelihood ratio of a 1-breakpoint recombinant to a direct non-recombinant descendant. This is in fact similar to the LARD approach (one sequence's evolutionary origin is evaluated by the likelihood ratio tests above) but in the LARD approach the query sequence is only compared against two other sequences whereas in RecombinHunt the query sequence is compared against many different lineages.

I understand that your paper does not do split decomposition. The LARD method does not do this either. The 1999 MBE paper (vol 16, no 3, pp.405-409, titles "Phylogenetic Evidence for Recombination in Dengue Virus") simply includes a split decomposition approach to help identify which parental sequences to test against. Look at the left column of page 406 of that paper. The long paragraph in the middle describes their likelihood approach which seems to be the same as the one in your paper. There's nothing wrong with that. But you should say where the approach comes from.

Thank you for this comment, which prompted us to better clarify the differences between our method and LARD and other classical approaches. We are absolutely convinced that proper credit must be clearly provided to the original approaches that first introduced likelihood-based metrics in the search for recombination events—we make our best efforts to address this properly, in the revised manuscript.

We now explicitly state that the RecombinHunt approach aims to detect a mosaic structure in viruses (in line with the terminology used in [4] and [5], from which our approach derives the idea of their Figures 1, as mentioned in the rebuttal of the first revision). This terminology has been added in the abstract and in the "Novelty" Section (Page 4, Line 79).

At the end of the "Introduction" Section, we added a paragraph to clarify which characteristics of our approach are inspired by previous valuable work in mosaic structure/recombination identification (see Pages 3-4, Lines 59-74):

"The approach implemented by RecombinHunt stems from a long-lasting tradition of statistical methods for the detection of intragenic recombination (started with [1]) – and is, in a way, related to substitution distribution models [2] – but differs in several key aspects.

First, RecombinHunt does not implement a triplet-based approach (such as RDP and 3SEQ) – where every candidate recombinant sequence is evaluated by extensive comparisons with all the potential

pairs of parents – but instead, it abstracts independent clusters of genomes as defined by a reference classification system/nomenclature in the form of a list of characterizing mutations. Subsequently, every candidate recombinant sequence is assessed by computing its similarity/dissimilarity with many existing lineages/groups of similar genome sequences.

Second, while previously established methods (such as LARD [3]) employ likelihood-based approaches to infer the most probable phylogenetic model and derive the evolutionary origin of a sequence (no-recombination, recombination, number of breakpoints) RecombinHunt does not reconstruct phylogenies but computes the likelihood of a collection of pre-defined designations/lineages and their combinations (recombinants) based on the mutations in the target sequence.

Third, although RecombinHunt identifies the most-likely candidate parents for a recombinant sequence by using an algorithm conceptually similar to the hypergeometric random walk described in [4, 5], unlike RecombinHunt these methods do not explicitly account for the frequency of each distinct point mutation, and are thereby bound to a totally different statistical framework.”

[1] Stephens, J.C., 1985. Statistical methods of DNA sequence analysis: detection of intragenic recombination or gene conversion. Molecular Biology and Evolution, 2(6), pp.539-556.

[2] Posada, D., Crandall, K.A. and Holmes, E.C., 2002. Recombination in evolutionary genomics. Annual Review of Genetics, 36(1), pp.75-97.

[3] Holmes, E.C., Worobey, M. and Rambaut, A., 1999. Phylogenetic evidence for recombination in dengue virus. Molecular biology and evolution, 16(3), pp.405-409.

[4] Boni, M.F., Posada, D. and Feldman, M.W., 2007. An exact nonparametric method for inferring mosaic structure in sequence triplets. Genetics, 176(2), pp.1035-1047.

[5] Lam, H.M., Ratmann, O. and Boni, M.F., 2018. Improved algorithmic complexity for the 3SEQ recombination detection algorithm. Molecular biology and evolution, 35(1), pp.247-251.

#2. The "Rejected Hypothesis Tree Problem": if you reject the hypothesis of no recombination, you can't build a tree. I appreciate that you now see this as a problem. And I liked parts of your response in the author rebuttal. You said:

"Our characteristic mutations are re-computed considering the mutations that are present in a parametric number of sequences of the lineage (75% for SARS-CoV-2); this results in an 'approximation' of the lineages derived from the tree, removing mutations introduced by recombinant sequences, wrong assignments to lineages, and other noise."

This might be a good approach to building an appropriate backbone tree, even when you have recombination, so that you can test recombinants against this backbone tree and all its lineages. But this method is not described anywhere in the ms. I appreciate that this is a very tall problem to solve, and that none of the lineage-assignment-recombination methods have solved this problem yet.

We are grateful to the reviewer for pointing out this aspect. We recognize that the method used to build the lineage characterizations was not sufficiently explained in the previous version of the manuscript. Thus, we added an additional paragraph in the Methods (see Pages 30-31, Lines 561-567): "Lineages are represented only considering the mutations that are present in a parametric number of sequences of the lineage (75% for SARS-CoV-2). These mutations are called characteristic mutations;

the list of characteristic mutations for a lineage is denoted as the lineage mutations-space. These data are used to estimate the baseline frequency of genomic variants across the complete collection of Pango lineages and in the SARS-CoV-2 genome. In the specific case of SARS-CoV-2 and the phylogenesis-based nomenclature that we use, this results in an ‘approximation’ of the lineages derived from the phylogenetic tree, removing mutations introduced by recombinant sequences, wrong assignments to lineages, and other noise.”

At the same time, at the beginning of “The approach” Section, we added (see Page 5, Lines 105-107): “Groups of genome sequences sharing the same label according to a structured nomenclature (i.e., lineages) are abstracted in the form of a list of characteristic mutations above a given frequency threshold (see Methods).”

If the other two referees are willing to drop point #2, I will also drop my criticism here, but this open problem in "recombination detection via lineage assignment" needs a paragraph in the discussion.

As suggested by the reviewer, we also added a paragraph that discusses benefits and limits of “recombination detection via lineage assignment” (see Pages 14-15, Lines 369-387): “RecombinHunt introduces a novel -big data-oriented- approach in the framework of likelihood-based methods for the detection of recombinant/mosaic genome structures. Two highly intertwined features define the key innovations of our method 1) usage of non-overlapping, potentially independent clusters of genome sequences, each characterized only by its prevalent mutations, for expressing the salient features of viral evolution; 2) a statistical framework built on the assessment of cumulative likelihood of a collection of characteristic mutations.

RecombinHunt does not directly depend on phylogenetic inference; however, the need for the definition of a structured nomenclature and/or a discrete set of designations with coherent features represents a fundamental prerequisite for the application of RecombinHunt. At the time being, most systems for the nomenclature/classification of human pathogens are based on phylogenies and, in this context, having an accurate phylogeny is critical for determining the correct frequency threshold for the identification of characterizing mutations of distinct groups. The ideal value of this parameter depends, for a given viral species, on the size of the sequence datasets and on the granularity of the employed classification, and hence might be different for different use cases, as discussed in the specificity section.

Further, we recognize that RecombinHunt does also present some methodological limitations. First, it can not detect recombination events supported by less than three mutations. Second, as we employ relative coordinates in the mutation-space rather than genomic coordinates (see also the gap-resolution procedure) the position of breakpoints is calculated with some degree of uncertainty. Nevertheless, none of these issues substantially impacts the ability of RecombinHunt to reliably construct a global SARS-CoV-2 recombination landscape.”

#3. In the new power analysis, you generated recombinant sequences and added noise by adding 0, 1, 2, or 3 mutations. This is not very many mutations, and any recombination method should be able to detect these recombinants. This is why you are getting 99% statistical power in this simulated data set.

*We agree that this analysis can be expanded by using higher mutational rates. We have repeated the analysis by using larger batches of sequences (500 instead of 250, to achieve better precision) and by simulating sequences with, respectively, 0, 3, 5, 10, 15, 20, and 30 mutations. The new results are described in the dedicated section (see Page 7, Lines 149-160) and the new Table 1a. It should be noted that the added 0-30 mutations are not to be intended on a genome-space of 29903 nucleotides, but on the target mutation-space (i.e., “genome positions with a mutation in the target” as we defined it on Page 5, Line 97). Then, in the revised manuscript (Page 7, Lines 154-156) we reported: “**Note that the generated SARS-CoV-2 genome sequences carry about 60 mutations compared to the reference Wuhan1 genome; hence, adding 3 mutations corresponds to inserting 5% noise, whereas 30 mutations to inserting 50% noise.**”*

For consistency with the sensitivity analysis, also the specificity analysis has been repeated with the larger and more noisy simulated datasets. The FPR spans from 0.6% when 1 random mutation is added, to 8.8% when 30 random mutations are added (to the pool of 65 initial mutations). The new results are described in the dedicated section (see Page 7, Lines 161-166) and the new Table 1b.

In a power analysis, you are supposed to compare RecombinHunt to GARD, 3SEQ, RDP, RIPPLES and show that RecombinHunt has similar or better power, similar or better specificity than these other methods. The specificity analysis on page 6 looks good to me.

I am very sorry to say this, and I really like the effort you have put into the revision, but a power analysis without other methods present is not good enough for a journal like Nature Communications. Please ask referees 2 and 3 if they agree with me.

We understand the reviewer’s concern. As GARD [1], 3SEQ [2] and RDP [3] are methods that are not conceived to be used with big data, therefore we defend that direct comparison with them is unfortunately practically infeasible.

Specifically, 3SEQ and RDP work directly on groups of three unlabelled sequences. This means that – reproducing our sensitivity analysis – would require 500 (child sequences) plus 1,000 (parent sequences) = 1,500 for each set of non-characteristic added mutations (0, 3, 5, 10, 15, 20, or 30), which would need to be combined (without repetition) in groups of three, each resulting into ~560 million combinations to be evaluated. The p-value obtained for the test would then need to be corrected for multiple testing by $1/\text{number_of_combinations}$. This becomes prohibitively expensive (unless the identity of the recombinant sample and its two parents were already known, which is not useful in practice).

We explored the possibility of running the analysis with GARD. Here, the complexity (as declared in the ref. [1]) is $O(L^s)$, where L is the length of the sequence and s is the number of mosaic segments. In the case of SARS-CoV-2, then $O(L^s) = 29903^2 \sim 9 \times 10^8$ for 1BP cases and $29903^3 \sim 2.7 \times 10^{13}$ for 2BP cases. This complexity must be considered for all the sequences in each of our sets. However, we noted that GARD’s provided public endpoint (<https://www.datamonkey.org/GARD/>) works on sequences with a length $\leq 12,000$ bases and rather small datasets (≤ 500 sequences); with such size, all jobs on the server take several days of computation. Hence, this implementation does not allow the computationally-intensive tests required for a comparison with RecombinHunt sensitivity analysis.

A summary of these reasons has been mentioned in the Discussion in the revised manuscript (see Page 16, Lines 404-406).

Finally, for what concerns RIPPLES, we provided a detailed comparison including all known SARS-CoV-2 recombinant lineages at the time of our analysis, and therefore we did not include this method in the sensitivity analysis.

[1] Kosakovsky Pond, S.L., Posada, D., Gravenor, M.B., Woelk, C.H. and Frost, S.D., 2006. GARD: a genetic algorithm for recombination detection. *Bioinformatics*, 22(24), pp.3096-3098.

[2] Lam, H.M., Ratmann, O. and Boni, M.F., 2018. Improved algorithmic complexity for the 3SEQ recombination detection algorithm. *Molecular biology and evolution*, 35(1), pp.247-251.

[3] Martin, D.P., Varsani, A., Roumagnac, P., Botha, G., Maslamoney, S., Schwab, T., Kelz, Z., Kumar, V. and Murrell, B., 2021. RDP5: a computer program for analyzing recombination in, and removing signals of recombination from, nucleotide sequence datasets. *Virus Evolution*, 7(1), p.veaa087.

Reviewer #2

(Remarks to the Author):

Summary

I appreciate the author's efforts in addressing reviewer comments in developing a specificity and sensitivity analysis. This manuscript is notably improved.

We thank the reviewer for reassessing our manuscript and providing positive feedback. Point-by-point replies are reported below.

Major Comments

I believe the authors may have misunderstood one of my prior major comments- I apologize for the miscommunication. I do not expect RecombinHunter to identify novel lineages from a phylogeny or generic sequence dataset; rather, the idea is to decide whether a small group of sequences identified via some other method is a recombinant of two established lineages. The authors do recognize how identification of donor/acceptor pairs can be part of this workflow in their response, and I would like to see this discussed and/or briefly exemplified in the work proper. I would particularly appreciate an explicit demonstration that RecombinHunter could have identified XBB and other lineages descended from a novel recombinant event accurately given only the data available at the time. The authors do of course analyze the efficacy of RecombinHunter as applied to all identified recombinant sequences and lineages in the wider dataset, but this is a slightly different problem from examining the theoretical historical performance of this approach in practice. This extension of the author's analysis, while not ultimately essential, would make it very clear how RecombinHunter is a valuable addition to the toolkit of genomic epidemiologists and lineage curators.

We thank the reviewer for the clarification. We understand the high interest of such a prediction-oriented application of RecombinHunt and thank you for the suggestion. The results of our additional analysis are described in the new section "Post hoc detection of recombinant lineages" (Page 11, Lines 263-279): "The original dataset analyzed in this manuscript represents a data freeze of the GISAID database as of April 1st, 2023. At that time the recombinant lineages XCA and XCB had not yet been designated by the Pango community. These lineages were first introduced in Pango, respectively on April 3rd, 2023 (XCA: BA.2.75 + BQ.1) (pango-issue) and April 14th, 2023 (XCB: BF.31.1 + BQ.1.10) (pango-issue).

Since our dataset already included 13 XCA-projected sequences (whose designation was subsequently changed to XCA) and 11 XCB-projected sequences (subsequently changed to XCB), we executed post hoc analyses on these two groups of sequences to evaluate the predictive power of RecombinHunt and its potential application for the early identification of recombinant sequences.

All of the 13 XCA-projected sequences were correctly labeled as recombinants of BA.2.75 and BQ.1 (12:1BP, 1:2BP) by RecombinHunt. Also the 11 XCB-projected sequences were all recognized as recombinants, in the majority of cases (8/11) with exactly the same designation as Pango (i.e., as

recombinants of BF.31.1 and BQ.1.10). In the remaining 3 cases, the method returned slightly different combinations of parent lineages, still compatible with the Pango designation of XCB. These results show that both XCA-projected and XCB-projected sequences would have been correctly classified as recombinant by RecombinHunt, and suggest that our method could contribute a significant advance in the early identification of candidate recombinant sequences/lineages.”

We must note that the XBB lineage had already been designated at the time when we downloaded the GISAID dataset (April 1st, 2023) – see Pango issue 1958, September 18, 2022. Hence, an equivalent analysis to the one made on XCA/XCB could not be performed.

However, 455 sequences (labeled as BA.2.10.1) and 46 sequences (labeled with other lineages) in that dataset are now assigned to the XBB designation (as of late January 2024). Out of these 501 sequences, 86% were labeled as recombinant by RecombinHunt (396 as 1BP; 35 as 2BP).

This highlights the fact that RecombinHunt would have recognized a larger proportion of XBB recombinant sequences in the original dataset, thus favoring a more accurate designation.

Minor Comments

I appreciate the authors clarifying that RIVET results were downloaded directly from the portal. I am mildly concerned about the apparent insensitivity of RIVET’s RIPPLES parameters, though it does avoid needing a curated lineage system as a starting point. Naturally, changes to RIVET are far beyond the scope of this work or the responsibility of the authors, though I would appreciate a brief statement that the insensitivity may be due to the specific configuration underlying the web portal rather than inherent to the statistical approach.

We understand the reviewer’s concern and agree that a specification is owed in our comparison section. We have added (see Page 12, Lines 294-295): “Since RIVET results were retrieved directly from its public endpoint we assumed them to reflect the optimal parameter configuration for this method.” With this, we also aim to clarify that, if other users will test the RIVET software with different parameters’ configurations, they may obtain different outcomes.

Reviewer #3

(Remarks to the Author):

The authors did a commendable job of addressing my major comments. They took my suggestions into account and implemented them effectively. The revisions made were well-thought-out and contributed significantly to the overall quality of their work. It is evident that the authors are dedicated to producing a high-quality piece, and I applaud their efforts.

Dear reviewer, we are very grateful for your positive assessment of our manuscript and for the time you dedicated to this work; we are making our best efforts to deliver a contribution that is valuable to the community.

REVIEWERS' COMMENTS

Reviewer #1 (Remarks to the Author):

It looks like the editors are going to lean towards acceptance. My technical and scientific reasoning tells me that this is a 'reject'.

I do believe that in a journal like Nature Communications the authors should show a power analysis that compares their approach to previous approaches. I don't believe the size of the data set is a concern.

I will leave the decision with the other two referees and the editor.

Reviewer #2 (Remarks to the Author):

I appreciate the author's efforts in responding to my concerns and clarifying the necessary points. I have no further concerns and recommend acceptance of the paper at this point.